# Compositional editing of extracellular matrices by CRISPR/Cas9 engineering of human mesenchymal stem cell lines

**Sujeethkumar Prithiviraj**[1,2], **Alejandro Garcia Garcia**[1,2], **Karin Linderfalk**[1,2],
**Bai Yiguang**[1,2,3], **Sonia Ferveur**[1,2], **Ludvig Nilsén Falck**[4],
**Agatheeswaran Subramaniam**[4], **Sofie Mohlin**[5], **David Hidalgo Gil**[1,2],
**Steven J Dupard**[1,2], **Dimitra Zacharaki**[1,2], **Deepak Bushan Raina**[6],
**Paul E Bourgine**[1,2]*

[1]Cell, Tissue & Organ Engineering Laboratory, BMC, Department of Clinical Sciences, Lund University, Lund, Sweden; [2]Wallenberg Centre for Molecular Medicine, Lund Stem Cell Centre, Lund University Cancer Centre, Lund University, Lund, Sweden; [3]Department of Orthopaedics, Nanchong Central Hospital, The Second Clinical Institute of North Sichuan Medical College Nanchong, Sichuan, China; [4]Division of Molecular Medicine and Gene Therapy, Lund Stem Cell Centre, Lund University, Lund, Sweden; [5]Division of Pediatrics, Clinical Sciences, Translational Cancer Research, Lund University Cancer Center at Medicon Village, Lund, Sweden; [6]The Faculty of Medicine, Department of Clinical Sciences Lund, Orthopedics, Lund, Sweden

**\*For correspondence:**
paul.bourgine@med.lu.se

**Competing interest:** The authors declare that no competing interests exist.

## eLife Assessment

The study presents a potentially **valuable** approach to genetically modify cells to produce extracellular matrices with altered compositions, termed cell-laid, engineered extracellular matrices (eECM). The evidence supporting the authors' conclusions regarding the utility of eECM for endogenous repair is **solid**, although there are some disagreements on the chondrogenicity of lyophilized constructs which was viewed as lacking robust evidence for endochondral ossification.

**Abstract** Tissue engineering strategies predominantly rely on the production of living substitutes, whereby implanted cells actively participate in the regenerative process. Beyond cost and delayed graft availability, the patient-specific performance of engineered tissues poses serious concerns on their clinical translation ability. A more exciting paradigm consists in exploiting cell-laid, engineered extracellular matrices (eECMs), which can be used as off-the-shelf materials. Here, the regenerative capacity solely relies on the preservation of the eECM structure and embedded signals to instruct an endogenous repair. We recently described the possibility to exploit custom human stem cell lines for eECM manufacturing. In addition to the conferred standardization, the availability of such cell lines opened avenues for the design of tailored eECMs by applying dedicated genetic tools. In this study, we demonstrated the exploitation of CRISPR/Cas9 as a high precision system for editing the composition and function of eECMs. Human mesenchymal stromal/stem cell (hMSC) lines were modified to knock out vascular endothelial growth factor (VEGF) and Runt-related transcription factor 2 (RUNX2) and assessed for their capacity to generate osteoinductive cartilage matrices. We report the successful editing of hMSCs, subsequently leading to targeted VEGF and RUNX2-knockout cartilage eECMs. Despite the absence of VEGF, eECMs retained full capacity to instruct ectopic endochondral ossification. Conversely, RUNX2-edited eECMs exhibited impaired

hypertrophy, reduced ectopic ossification, and superior cartilage repair in a rat osteochondral defect. In summary, our approach can be harnessed to identify the necessary eECM factors driving endogenous repair. Our work paves the road toward the compositional eECMs editing and their exploitation in broad regenerative contexts.

## Introduction

Extracellular matrices (ECMs) are complex networks of proteins not only providing tissue structural and mechanical support, but also acting as growth factor storing and presenting entities (*Bonnans et al., 2014*). As such, ECMs are receiving increasing attention in tissue engineering as templates capable of guiding homeostasis, remodeling, and regenerative processes (*Hussey et al., 2018*; *Fattahi et al., 2022*; *Mouw et al., 2014*).

ECMs can be derived from native tissue or organs (native ECMs [nECMs]) by applying a decellularization step which effectively removes the cellular fraction. The resulting nECMs are commonly used as biomaterials in regenerative medicine (*Assunção et al., 2020*; *Hinderer et al., 2016*), offering a natural biocompatible structure. However, they suffer from substantial batch-to-batch variation with their properties being largely affected by the tissue source and the decellularization process (*Assunção et al., 2020*; *Hinderer et al., 2016*). Importantly, while displaying a high level of biological complexity and fidelity, the composition of nECMs cannot be tailored to specific needs.

A valuable option emerged from recent advances in bioengineering which led to the development of synthetic ECMs (sECMs). These tunable materials are predominantly composed of biopolymers such as poly(lactic acid), poly(glycolic acid), poly(lactic co-glycolic acid), and poly(ethylene glycol), which can be functionalized with bioactive substances such as peptides, growth factors, or enzymes (*Fattahi et al., 2022*; *Tang et al., 2021*; *Gao et al., 2017*). The possible modulation of sECM's composition and chemistry offers precise control over their mechanical properties, degradability, and temporal release of instructive molecules toward improved tissue repair. Despite holding great promises, their structure and function remain simplified compared to their native counterpart (*Filippi et al., 2020*; *Reddy et al., 2021*).

With the aim of combining high biological fidelity and design flexibility, engineered ECMs (eECMs) have become a credible alternative. Those result from the exploitation of stem/progenitor populations capable of depositing ECMs under specific in vitro culture conditions (*Decaris and Leach, 2011*; *Harvestine et al., 2018*; *Hoch et al., 2016*; *Haumer et al., 2018*), which can be subsequently isolated from the cellular fraction and exploited as a cell-laid product. Early studies were conducted using eECM derived from primary cells, exhibiting variable performance associated with the inter-donor variability (*Sadr et al., 2012*; *Prewitz et al., 2013*).

Most recently, we demonstrated the possibility to standardize the production of eECMs through the engineering of dedicated human mesenchymal stromal/stem cell (hMSC) lines (*Bourgine et al., 2017*). Precisely, the mesenchymal sword of Damocles bone morphogenetic type-2 (MSOD-B) line offered the generation of eECMs in the form of human cartilage templates (*Pigeot et al., 2021*). Those cartilage templates can be further devitalized, lyophilized, and stored as an *off-the-shelf* tissue, retaining remarkable bone formation capacity by instructing endochondral ossification (*Grigoryan et al., 2022*).

The exploitation of cell lines as an unlimited cell source for eECM generation offers unprecedented standardization in graft production and performance. Meanwhile, it also consists of a robust cellular platform facilitating their further genetic engineering, to achieve the production of eECMs tailored in composition and function. Toward this objective, CRISPR/Cas9 appears as a versatile and efficient gene-editing tool. While this technology has been extensively used for genetic screening or disease modeling (*Golchin et al., 2020*; *Hazrati et al., 2022*; *Geurts and Clevers, 2023*), the possibility to edit hMSCs and study the compositional impact on deposited eECMs has not been investigated so far.

In this study, we aim at demonstrating that CRISPR/Cas9 can be applied as an editing tool for the engineering of custom eECMs. As proof-of-principle, our strategy relies on the CRISPR/Cas9-guided editing of the MSOD-B line for knocking out vascular endothelial growth factor (VEGF) and Runt-related transcription factor 2 (RUNX2), respectively, a key pro-angiogenic protein and transcription factor critically involved in the endochondral ossification pathway. The resulting hMSC lines will be

evaluated for their capacity to generate hypertrophic cartilage in vitro and instruct tailored skeletal tissue formation in vivo as cell-free and lyophilized templates. If successful, our study is expected to validate the use of CRISPR/Cas9 for eECMs edited in composition, while providing a novel platform toward decoding the necessary molecular signals driving effective tissue repair.

## Results

### CRISPR/Cas9 editing of hMSC lines lead to efficient VEGF knockout in cartilage tissues

VEGF is a known master regulator of angiogenesis (*Hu and Olsen, 2016*) and a key mediator of endochondral ossification. When reaching hypertrophy, chondrocytes highly express VEGF, prompting vasculature invasion and subsequent osteoprogenitor recruitment. This was proven to be essential for cartilage template remodeling into bone and bone marrow (*Duda et al., 2023*). However, whether VEGF is a requirement to instruct ectopic endochondral ossification remains to be investigated. Here, we thus first aimed at engineering a hMSC line CRISPR/Cas9-edited for VEGF knockout and evaluate the corresponding impact on cartilage and endochondral bone formation (*Figure 1A*). To this end, we exploited the MSOD-B as human cell line previously demonstrated as capable of endochondral ossification (*Pigeot et al., 2021*; *Figure 1—figure supplement 1*). Specific guide RNAs (gRNAs) targeting different regions of the VEGF gene but conserved across all isoforms of VEGF were designed based on a previously established protocol (*Ran et al., 2013*) toward modifying the MSOD-B line. Three gRNAs targeting exon 1 (VEGF_1.1, VEGF_1,2, VEGF_1.3), one targeting exon 2 (VEGF_2.1), and one targeting exon 8 (VEGF_8.1) were designed (*Figure 1B*, *Figure 1—figure supplement 1*). These gRNAs were cloned into the pU6-(BbsI)_ CBh-Cas9-T2A-mCherry vector encoding the *Streptococcus pyogenes* Cas9 (SpCas9) machinery. Following transfection in MSOD-B cells, single cell clones were sorted based on the transient mCherry expression and expanded for characterization (*Figure 1A*). Out of 163 single clones, 14 could be successfully expanded (8.5 %). From these clones, five were retrieved from the gRNA 1.1, six from gRNA 1.3, and three from gRNA 8.1. In order to assess a successful editing and directly correlate it to a knockout of VEGF, an ELISA was performed to measure the concentration of VEGF in the supernatant of expanded MSOD-B clones (*Figure 1C*). In non-edited cells (MSOD-B, control), 5000 pg/mL were secreted and detected by ELISA. From the successfully expanded clones, most of them showed reduced but non-abolished secretion of VEGF. However, two clones exhibited undetectable levels of VEGF, both resulting from the gRNA targeting the Exon1 of the VEGF gene. These clones were defined as MSOD-BΔV1 and MSOD-BΔV2 and further characterized for their capacity to form cartilage.

To this end, clones were expanded and seeded on collagen scaffold and induced for 3 weeks toward chondrogenic differentiation followed by lyophilization of the tissues. As anticipated with the MSOD-B tissues, histological assessment revealed the formation of a collagenous matrix with cartilage features (Safranin O, *Figure 1D*). Similarly, a successful chondrogenic differentiation and deposition of glycosaminoglycans could be observed in the MSOD-BΔV1 (*Figure 1D*) constructs. A quantitative assessment (Blyscan assay) confirmed the content in glycosaminoglycans in MSOD-B and VEGF-edited clones falling in the same concentration range although a lower amount was detected in the latest group (*Figure 1E*). Importantly, VEGF quantification in corresponding pellets validated the successful knockout of the protein, barely detectable in the MSOD-BΔV1 and ΔV2 tissues (47.64 pg/ pellet in edited clones vs 552.7 pg/pellet in MSOD-B, *Figure 1F*).

To further examine the potential impact of VEGF-editing on tissue formation, we performed immunostaining analysis of the engineered cartilage constructs. Confocal microscopy revealed strong deposition of Collagen Type I (COL1) and Collagen Type X (COLX) in all samples, characteristic of mature hypertrophic cartilage tissues (*Figure 1G*). However, while the immunostaining revealed the VEGF deposition in the MSOD-B ECM, the protein could not be detected in the MSOD-BΔV1 samples (*Figure 1G*).

Taken together, this data indicates the successful generation of MSOD-B lines knocked out in VEGF. The MSOD-BΔV1 clones exhibited cartilage formation capacity with minimal level of VEGF. This validates the use of CRISPR/Cas9 as a precision tool to edit the composition of cartilage tissue.

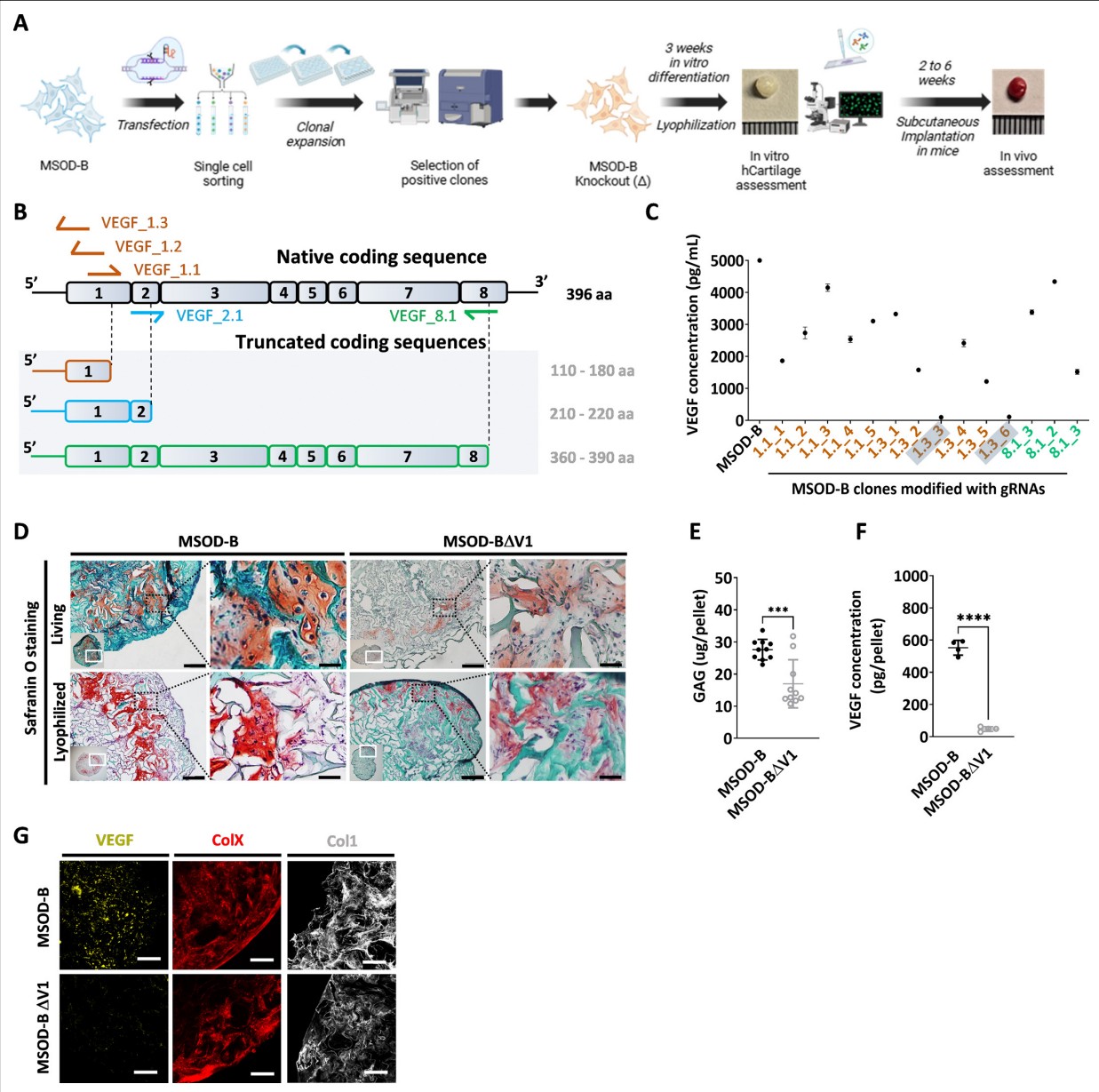

**Figure 1.** CRISPR/Cas9 editing of mesenchymal cell lines lead to efficient vascular endothelial growth factor (VEGF) knockout in cartilage tissues. (**A**) Experimental scheme depicting the generation of CRISPR/Cas9-edited mesenchymal sword of Damocles bone morphogenetic type-2 (MSOD-B) lines, and the subsequent in vitro and in vivo tissue formation assessment. (**B**) Overview of the native human VEGF coding sequence composed of eight exons. Designed guide RNAs (gRNAs) and their targeted binding sites are illustrated, as well as the corresponding expected impact on the coding sequence. gRNAs targeting exon 1 (orange) disrupt translation initiation and inhibit protein expression, gRNA targeting exon 2 (blue) disrupts VEGF receptor binding, and gRNA targeting exon 8 (green) alters the C-terminal sequence and represses activation of protein. (**C**) ELISA-based quantitative analysis of VEGF protein content in cell culture supernatant from expanded single cell colonies. From all clones, only two had no detectable level of VEGF (1.3_3 and 1.3_6). These clones were subsequently defined as MSOD-BΔV1 and MSOD-BΔV2. (**D**) Histological assessment of living and lyophilized in vitro differentiated constructs using Safranin O staining (scale bars=100 µm and 20 µm for magnified areas). Both the MSOD-B and MSOD-BΔV1 displayed glycosaminoglycans (GAG) (orange to reddish in Safranin O), indicating successful cartilage formation. Left bottom inserts show the whole tissue section. (**E**) Quantitative assessment of the total GAG content in MSOD-B and MSOD-BΔV1 in vitro differentiated constructs, post-lyophilization. Unpaired t-test, n=10 biological replicates, ***p<0.001. (**F**) ELISA-based quantitative assessment of VEGF protein in in vitro differentiated constructs, post-lyophilization. Unpaired t-test, n=3–4 biological replicates, ****p<0.0001. (**G**) Immunofluorescence images of MSOD-B and MSOD-BΔV1 tissues, post-lyophilization. Displayed images consist of 3D-stacks from 80- to 100-µm-thick sections, stained for VEGF (yellow), Collagen Type X (COLX, red), and Collagen Type I (COL1, gray). A clear reduction in the VEGF signal could be observed in the MSOD-BΔV1 tissues, indicating a successful VEGF knockdown (scale bars=80 µm). All error bars in the figures indicate the standard deviation (SD). Panel A was created with BioRender.com.

*Figure 1 continued on next page*

*Figure 1 continued*

The online version of this article includes the following figure supplement(s) for figure 1:

**Figure supplement 1.** Implemented genetic elements of the mesenchymal sword of Damocles vascular endothelial growth factor (VEGF) (MSOD-BΔV) cells consisting of the human telomerase reverse transcriptase (hTERT), an inducible caspase 9 (iCaspase) death system, the bone morphogenetic protein type-2 and VEGF knockdown and the overview of the human VEGF exon structure, nucleotide sequences of targeting guide RNAs (gRNAs) and their corresponding amino acid sequences. Panel A was created with BioRender.com.

## VEGF knockout cartilage tissues retain bone remodeling capacity despite reduced early-stage vascularization

To evaluate the functional impact of VEGF knockout in engineered constructs, we assessed both the angiogenic and bone formation assays. Since the two clones exhibited similar in vitro tissue formation capacity, only the MSOD-BΔV1 was selected for further performance assessment.

We first performed the chorioallantoic membrane (CAM) assay as ex vivo assessment of angiogenic potential (*Figure 2A*). This assay provides insights into the biological activity and neovascularization potential of each construct, through quantitative evaluation of vascular density around engineered grafts. After 4 days, we observed an extensive vessel formation in the periphery of both MSOD-B and MSOD-BΔV1 lyophilized eECMs (*Figure 2B*). Quantification of vascular densities (*Callewaert et al., 2023*) suggested a reduced vessel formation in MSOD-BΔV1 samples but without reaching significance (*Figure 2C*, *Figure 2—figure supplement 1*). Thus, the CAM assay indicated that both MSOD-B and MSOD-BΔV1 cartilages retain angiogenic potentials.

We next investigated in vivo the impact of the VEGF knockout on the graft capacity to undergo endochondral ossification, by implanting cartilage grafts subcutaneously in an immunodeficient mouse model. Importantly, prior to implantation, tissues were lyophilized using a preestablished protocol (*Pigeot et al., 2021*). This allowed us to assess the performance of the generated tissue itself, in line with the idea of developing off-the-shelf substitutes. Samples were extracted after 2 and 6 weeks as early and late development time points (*Figure 2D*). After 2 weeks in vivo, we first evaluated the angiogenic potential of tissues by quantitative confocal microscopy imaging of 100-µm-thick sections stained for CD31, a well-defined vascular marker. This allowed us to evidence a dense vascular network in MSOD-B constructs, covering the entirety of the grafts (*Figure 2E*). Instead, MSOD-BΔV1 displayed a more limited vascularization, predominantly at the tissue periphery (*Figure 2E*). Quantification confirmed these observations with MSOD-B exhibiting a significantly higher volume of vessels per total section volume (0.523 µm$^3$ vs 0.231 µm$^3$ for MSOD-B and MSOD-BΔV1, respectively, *Figure 2F*).

Using microtomography, we further assessed the amount of mineralized tissue formed in a temporal fashion. At 2 weeks, the formation of a cortical ring could already be observed in both MSOD-B and MSOD-BΔV1 tissues (*Figure 2G*). Quantifications revealed a similar bone/mineralized volume normalized to the total volume of the graft (BV/TV, *Figure 2H*, *Figure 2—figure supplement 1*, *Figure 2*, and *Figure 3*) with 17% and 16% in MSOD-B and MSOD-BΔV1 samples, respectively. At the 6-week time point, trabecular structures could be observed in reconstructed 3D scans from both groups (*Figure 2G*). Remarkably, histological analysis confirmed the similar development of the tissues with bone formation (Masson's trichrome) already at 2 weeks postimplantation and minimal remnants of cartilage (Safranin O) (*Figure 2I*). At 6 weeks both samples remodeled into fully mature bone organs, characterized by the presence of cortical and trabecular structures as well as a bone marrow tissue filling the cavity (*Figure 2I*).

These results indicate that the absence of VEGF in cartilage tissue can delay the early vascularization of MSOD-BΔV samples. However, this did not prevent nor impact the remodeling of the lyophilized grafts into bone and bone marrow tissues indicating that VEGF is nonessential in order to efficiently instruct endochondral ossification.

## RUNX2 knockdown does not prevent chondrogenic differentiation but impairs hypertrophy

We next investigated whether the composition and thus function of a graft could be modified by editing transcriptional factors involved in mesenchymal cell differentiation. Using CRISPR/Cas9, we thus targeted the RUNX2, a known master regulator particularly important for chondrocyte differentiation and hypertrophy (*Otto et al., 1997*; *Komori et al., 1998*). We designed gRNAs targeting

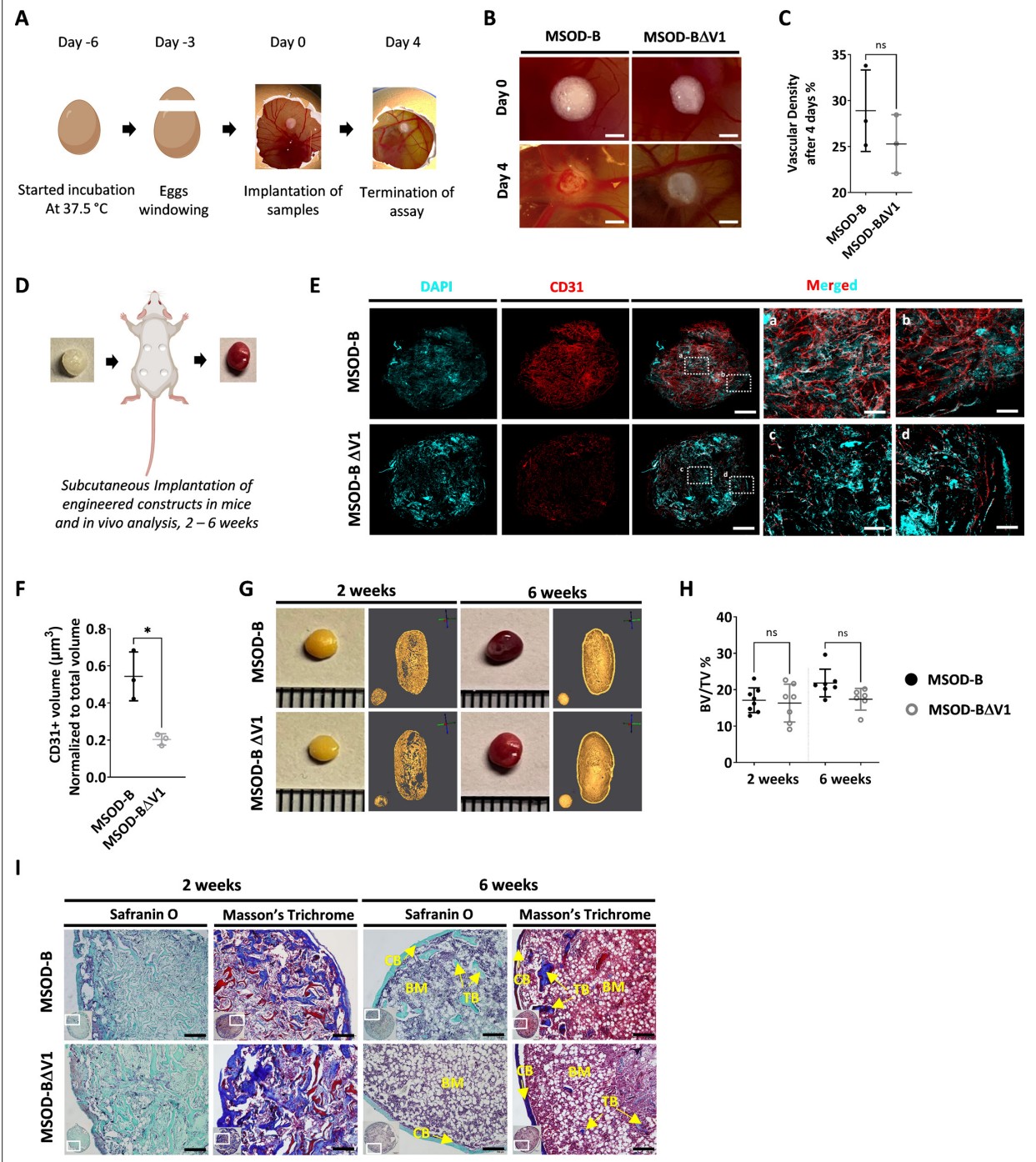

**Figure 2.** Vascular endothelial growth factor (VEGF) knockout cartilage tissues retain bone remodeling capacity despite reduced early-stage vascularization. (**A**) Experimental scheme of the chorioallantoic membrane (CAM) assay, for ex vivo evaluation of angiogenic potential. (**B**) Macroscopic comparison of in vitro mesenchymal sword of Damocles bone morphogenetic type-2 (MSOD-B) and MSOD-BΔV1 constructs at day 0 (day of implantation) and day 4 (4 days postimplantation), illustrating robust de novo vessel formation perfusing the tissues (scale bars at 1 mm). (**C**) Quantitative analysis of vascular density in MSOD-B and MSOD-BΔV1. Vascular densities were quantified from macroscopic images obtained on day 4 using ImageJ. Unpaired t-test, n=3–4 biological replicates, n.s.=not significant. (**D**) Overview of the subcutaneous implantation procedure in mice and subsequent in vivo evaluation of vascularization from 2 to 6 weeks postimplantation. (**E**) Immunofluorescence images of MSOD-B and MSOD-BΔV1 tissues 2 weeks post-in vivo implantation. Displayed images consist of 3D-stacks from 80 to 100-µm-thick sections, vessels stained with mouse CD31 (red) and nuclei with DAPI (cyan) (scale bars (for DAPI, CD31 and MERGED) at 500 µm except for magnified white inserts at 80 µm). Box 'a' and 'c' display the periphery whereas Box 'b' and 'd' show the central region of MSOD-B and MSOD-BΔV1 constructs respectively. A reduction in tissue vascularization is observed in MSOD-BΔV1 samples. (**F**) Quantitative analysis of the CD31 signal using an isosurface-based strategy (IMARIS software). Unpaired t-test,

*Figure 2 continued on next page*

*Figure 2 continued*

n=3–4 biological replicates, *p<0.05. (**G**) Representative macroscopic and microtomography images of in vivo constructs retrieved at 2 and 6 weeks postimplantation. (**H**) Microtomography-based quantification of the sample's bone/ mineralized volume over their total volume (ratio). No significant differences between MSOD-B and MSOD-BΔV1 could be observed (BV: bone volume, TV: total volume). Ordinary one-way ANOVA, n=8 biological replicates, n.s.=not significant. (**I**) Histological analysis of in vivo tissues using Safranin O and Masson's trichrome stains, at 2 (2W) and 6 weeks (6W) postimplantation. Both sample types underwent full remodeling into a bone organ after 6 weeks, with presence of bone structures and a bone marrow compartment (scale bars=200 µm). CB – cortical bone; BM – bone marrow; TB – trabecular bone. All error bars in the figures indicate the standard deviation (SD).

The online version of this article includes the following figure supplement(s) for figure 2:

**Figure supplement 1.** Sequential processing of a vascular tissue section using Q-VAT, depicting the original image, binary mask creation, and vessel segmentation for quantitative vascular density analysis.

**Figure supplement 2.** Microtomography analysis (bone/mineralized volume) showed significant differences between in vivo constructs at both 2 weeks and 6 weeks time points (one-way ANOVA, n=3, **p<0.01, n.s.=not significant).

**Figure supplement 3.** Microtomography analysis (total volume) did not show significant difference between in vivo constructs at either 2 weeks or 6 weeks time points (one-way ANOVA, n=3, n.s.=not significant) (BV: bone volume, TV: total volume).

the coding regions involved in cell signaling integration, activation, and inhibition domain which are conserved in both isoforms of RUNX2 (*Figure 3A*, *Figure 3—figure supplement 1*).

One gRNA was designed for targeting exon 2 (RUNX2_2.1), one targeting exon 5 (RUNX2_5.1), two targeting exon 6 (RUNX2_6.1 and RUNX2_6.2), and one targeting exon 8 (RUNX2_8.1). Similar to the VEGF setup, gRNAs were transfected in MSOD-B cells together with a pU6-(BbsI)_CBh-Cas9-T2A-mCherry plasmid to ensure transfection efficiency and single cell sorting of positive clones. Out of 385 single clones, 62 could be successfully expanded, corresponding to a 16.1% efficiency. From these clones, five were derived from the RUNX2_5.1 gRNA, 18 from RUNX2_6.1, 15 from RUNX26.2, and 24 from RUNX2_8.1.

To identify successfully edited clones, we first screened for those exhibiting a decrease in their RUNX2 protein expression using intracellular flow cytometry. Among the 62 clones, 17 displayed a reduced RUNX2 pattern as compared to the MSOD-B control (*Figure 3B*, *Figure 3—figure supplement 2*) and were further sent for sequencing analysis. This led to the identification of two successfully edited clones, demonstrating a point mutation in the exon 6 of RUNX2 gene (*Figure 3—figure supplement 3*). To confirm the knockout impact on the transcription factor structure, a western blot analysis was conducted on cellular extracts of in vitro cultured cells. MSOD-B cells exhibited intact RUNX2 proteins of 52–62 kDa (*Kim et al., 2020*; *Hattori et al., 2022*; *Figure 3C*). Instead, the RUNX2-edited cells displayed truncated versions of 32–42 kDa, consistent with the expected mRNA shortening.

These two clones were defined as MSOD-BΔR1 and MSOD-BΔR2 and further assessed for their ability to form cartilage in vitro. Following 3D chondrogenic differentiation, samples were lyophilized and processed for histological analysis. The presence of cartilage structures embedded in a collagenous matrix could be observed in all MSOD-B, MSOD-BΔR2 (*Figure 3D*), and MSOD-BΔR1 tissues (*Figure 3—figure supplement 4*). This was confirmed quantitatively using the Blyscan assay, with detectable glycosaminoglycans in all groups (*Figure 3E*, *Figure 3—figure supplement 5*). Using immunostaining combined with quantitative confocal microscopy, the possible impact of RUNX2-editing on tissue hypertrophy was further investigated. First, Collagen Type II (COL2) staining confirmed the distribution of cartilage matrix across MSOD-B, MSOD-BΔR1, and MSOD-BΔR2 samples (*Figure 3F*, *Figure 3—figure supplement 6*). However, we observed a clear distinction in the Collagen Type X (COLX) expression pattern, a specific marker of hypertrophy which was hardly detectable in MSOD-BΔR tissues (*Figure 3F*, *Figure 3—figure supplement 6*). Subsequent quantification confirmed a significant reduction of COLX volume in MSOD-BΔR1 (0.236 µm$^3$) and MSOD-BΔR2 (0.182 µm$^3$) compared to MSOD-B (0.81 µm$^3$) (*Figure 3G and H*, *Figure 3—figure supplements 7 and 8*, and *Figure 3—figure supplement 9*).

To further characterize the functional impact of the RUNX2 knockdown, the MSOD-B and MSOD-BΔR1 osteogenic differentiation capacity was assessed in vitro. After 3 weeks of culture in osteogenic medium (or expansion medium, control), Alizarin Red staining revealed as a marked absence of mineralization in MSOD-BΔR1 culture, while in stark contrast with MSOD-B cells (*Figure 3I*). Quantitative polymerase chain reaction (qPCR) analysis indicated a lower expression of COLX and ALPL in MSOD-BΔR1 (*Figure 3J*), key osteogenic-associated genes, of interest, the expression level

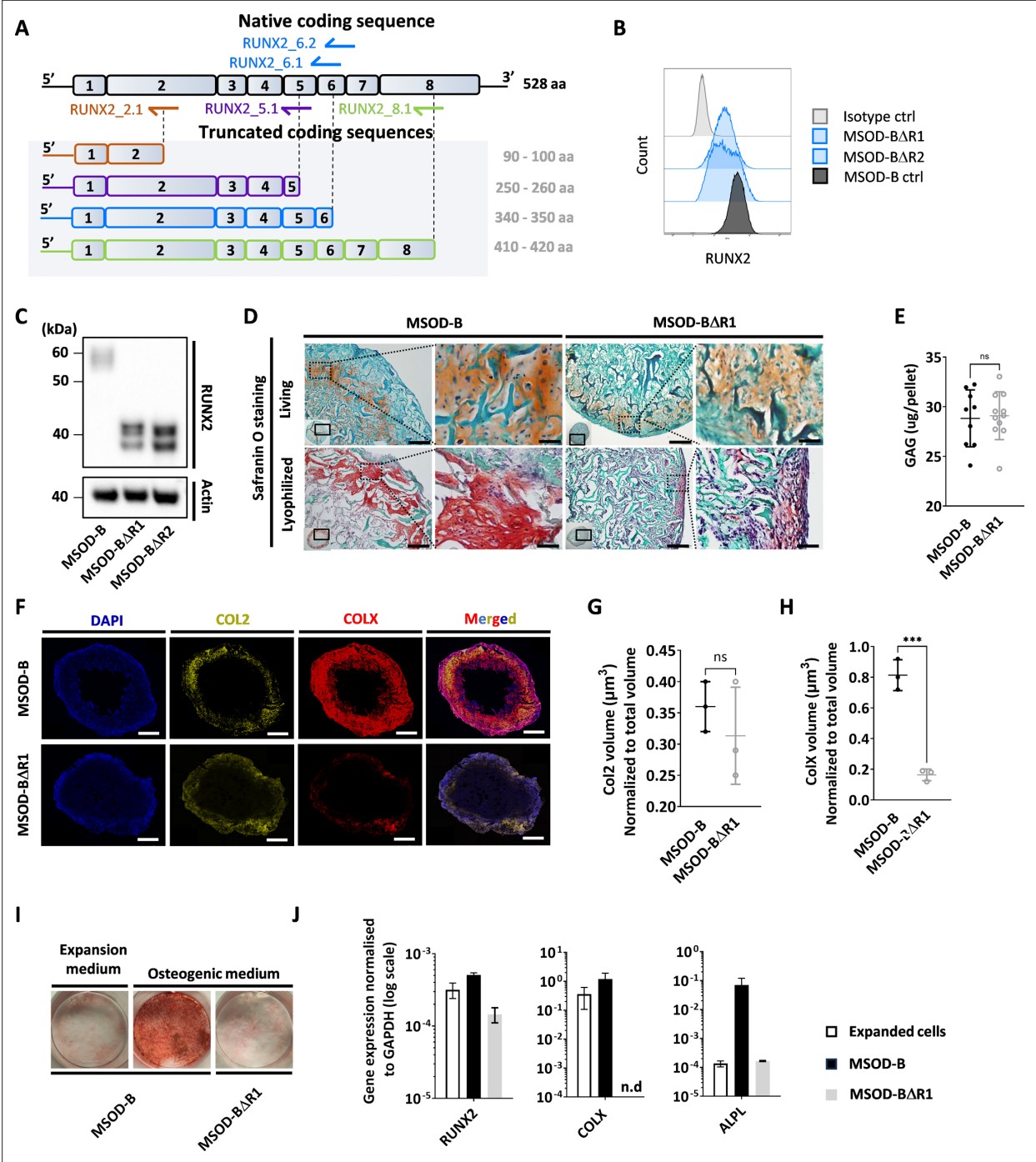

**Figure 3.** Runt-related transcription factor 2 (RUNX2) knockout does not prevent chondrogenic differentiation but impairs hypertrophy. (**A**) Overview of the human RUNX2 coding sequence comprising eight exons. Guide RNAs (gRNAs) and their corresponding expected protein structure. The gRNA targeting exon 2 (orange) disrupts the DNA binding domain, gRNA targeting exon 5 (violet) disrupts nuclear translocation, gRNAs targeting exon 6 (blue) disrupt the transcriptional activation domain, and gRNAs targeting exon 8 (green) disrupt the nuclear matrix targeting signal and repress the protein function. (**B**) Intracellular flow cytometry for RUNX2 detection in mesenchymal sword of Damocles bone morphogenetic type-2 (MSOD-B) and RUNX2-edited clones. A clear protein reduction could be observed in the 6.1_1 and 6.1_23 clones. (**C**) Western blot analysis of RUNX2 in cultured MSOD-B and RUNX2-edited cells. The genetic editing of RUNX2 is confirmed by the detection of the truncated proteins. Actin is used as a control to normalize the protein content. (**D**) Histological analysis of living and lyophilized in vitro differentiated constructs using Safranin O staining (scale bars=100 μm and 20 μm for magnified areas), indicating the presence of cartilage matrices. (**E**) Quantitative assessment of the total GAG content in corresponding in vitro generated lyophilized tissues. Unpaired t-test, n=10–11 biological replicates **p<0.01. (**F**) Immunofluorescence images of MSOD-B and MSOD-BΔR1 lyophilized tissues. Displayed images consist of 3D-stacks from 80 to 100 μm thick sections, stained for DAPI (blue), Collagen

*Figure 3 continued*

Type II (COL2, yellow), and Collagen Type X (COLX, red). A clear reduction in the COLX signal could be observed in the MSOD-BΔR1 tissues, indicating impaired hypertrophy (scale bars=500 µm). (**G**) Isosurface-based quantification of the COL2 immunofluorescent signal using the IMARIS software. No significant difference between groups confirms the retention of cartilage formation in RUNX2-edited constructs. Unpaired t-test, n=3 biological replicates, n.s.=not significant. (**H**) Isosurface-based quantification of the COLX immunofluorescent signal using the IMARIS software, confirming the disruption of hypertrophy in the MSOD-BΔR1 constructs. Unpaired t-test, n=3, ***p<0.001. (**I**) Alizarin Red staining evidencing a lack of mineralization in the MSOD-BΔR1 culture compared to the MSOD-B. (**J**) Quantitative polymerase chain reaction (PCR) analysis displaying the relative expression levels of osteogenesis-related genes: RUNX2, COL1, and ALPL. The expression is normalized to GAPDH as housekeeping gene. n.d.=not detected. All error bars in the figures indicate either the standard deviation (SD) or the standard error of the mean (SEM); the specific metric used is consistent within each figure.

The online version of this article includes the following source data and figure supplement(s) for figure 3:

**Source data 1.** *Figure 3C* annotated western blot analysis of Runt-related transcription factor 2 (RUNX2) editing.

**Source data 2.** *Figure 3C* annotated western blot analysis of Runt-related transcription factor 2 (RUNX2) editing.

**Figure supplement 1.** Implemented genetic elements of the mesenchymal sword of Damocles Runt-related transcription factor 2 (RUNX2) (MSOD-BΔR) cells consisting of the human telomerase reverse transcriptase (hTERT), an inducible caspase 9 (iCaspase) death system, the bone morphogenetic protein type-2 and RUNX2 knockdown and an overview of human RUNX2 exon structure, nucleotide sequences of targeting guide RNAs (gRNAs) and its corresponding amino acid sequence. Partly created with BioRender.com.

**Figure supplement 2.** Intracellular flow cytometry for Runt-related transcription factor 2 (RUNX2) detection in mesenchymal sword of Damocles bone morphogenetic type-2 (MSOD-B) and RUNX2-edited clones.

**Figure supplement 3.** Sanger sequencing results of DNA extracted from mesenchymal sword of Damocles (MSOD)-BΔR samples and the modifications compared to MSOD-B DNA sequence.

**Figure supplement 4.** Histological analysis of in vitro constructs from using Safranin O and Masson's trichrome staining displayed the presence of glycosaminoglycans (GAG) and collagen content, respectively, indicating the presence of cartilage formation in mesenchymal sword of Damocles (MSOD)-BΔR2 (scale bars=200 µm).

**Figure supplement 5.** Quantitative assessment of the total GAG content confirms the presence of cartilage formation in both constructs thus confirming Runt-related transcription factor 2 (RUNX2) knockout in human mesenchymal stromal/stem cells (hMSCs) (mesenchymal sword of Damocles [MSOD]-BΔR2) retain cartilage formation (unpaired t-test, n=3, n.s.=not significant).

**Figure supplement 6.** Reduction of Runt-related transcription factor 2 (RUNX2) expression in cartilage tissues lead to reduced hypertrophy.

**Figure supplement 7.** Quantitative analysis of the immunofluorescent staining sections using IMARIS software displayed significant difference in the expression of COL2, demonstrating reduction of cartilage formation in knockout constructs (mesenchymal sword of Damocles [MSOD]-BΔR2) (unpaired t-test, n=3, *p<0.05).

**Figure supplement 8.** Quantitative analysis of the immunofluorescent staining sections using IMARIS software displayed a significant reduction in expression of COLX, confirming the disruption of hypertrophy in knockout constructs (unpaired t-test, n=3, **p<0.01).

**Figure supplement 9.** Representative isosurface images of immunofluorescent staining sections constructed using IMARIS software.

of RUNX2 was only partially decreased in MSOD-BΔR1, in line with the truncated mRNA reducing but not abrogating qPCR primers binding probability.

In summary, we here report the successful RUNX2 knockout in MSOD-B lines using CRISPR/Cas9. RUNX2 did not impair chondrogenesis but prevented the tissue hypertrophy as well as the osteogenic potential of the cells.

## RUNX2 knockout in cartilage tissues disrupts effective ectopic bone formation

To assess the corresponding impact of RUNX2-edited cartilages on bone formation, MSOD-B tissues (as control) and MSOD-BΔR1 were lyophilized and implanted subcutaneously in immunodeficient mice for 2–6 weeks. From microtomography image reconstructions, we observed an evident reduction of mineralization in MSOD-BΔR1 and compared to MSOD-B (*Figure 4A*) already after 2 weeks. This qualitative and visual difference in mineralization persisted after 6 weeks in vivo (*Figure 4A*). Subsequent quantifications confirmed these observations with a clear reduction in BV/TV as early as week 2 (16.46% in MSOD-B and 7.01% in MSOD-BΔR1) and persisting at week 6 (20.03% in MSOD-B and 4.25% in MSOD-BΔR1) (*Figure 4B*, *Figure 4—figure supplement 1*, and *Figure 4—figure supplement 2*). Histological analysis revealed an early bone formation at 2 weeks in both samples, but to a lower extent in the MSOD-BΔR1 group (*Figure 4C*). As anticipated, MSOD-B tissues underwent full

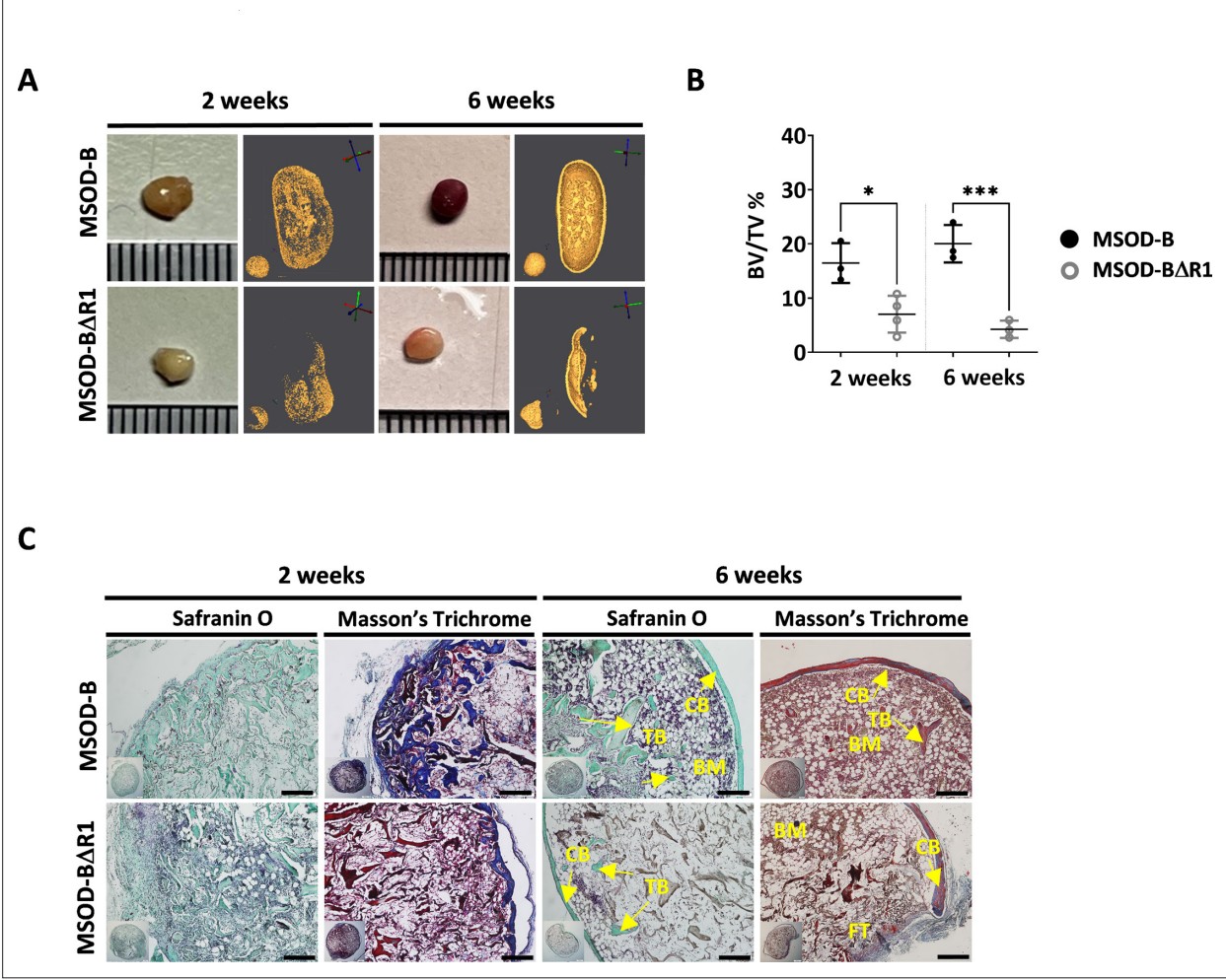

**Figure 4.** Runt-related transcription factor 2 (RUNX2) knockout in cartilage tissues disrupts effective ectopic bone formation. (**A**) Representative macroscopic and microtomography images of in vivo constructs retrieved at 2 and 6 weeks postimplantation. (**B**) Microtomography-based quantification of the sample's bone/mineralized volume over their total volume (ratio) (BV: bone volume, TV: total volume). A marked difference is observed as early as 2 weeks, with a clear lower mineral content in mesenchymal sword of Damocles (MSOD)-BΔR1 samples. Ordinary one-way ANOVA, n=3 biological replicates, *p<0.05, ***p<0.001. (**C**) Histological analysis of in vivo constructs using Safranin O and Masson's trichrome stains. After 2 weeks (2W), a higher bone formation is already evident in the MSOD-B control group. The MSOD-BΔR1 samples explanted after 6 weeks (6W) displayed presence of cortical and trabecular bone, but also large amount of fibrous tissue indicating an incomplete remodeling (scale bars=200 µm). The error bars in the figures indicate the standard deviation (SD).

The online version of this article includes the following figure supplement(s) for figure 4:

**Figure supplement 1.** Microtomography analysis (bone/mineralized volume) showed very high significant difference between in vivo constructs at both 2 weeks and 6 weeks time points (one-way ANOVA, n=3, ***p<0.001, ****p<0.0001).

**Figure supplement 2.** Microtomography analysis (total volume) showed significant difference between in vivo constructs at 2 weeks time point but no significant difference at 6 weeks time point (one-way ANOVA, n=3, n.s.=not significant) (BV: bone volume, TV: total volume).

**Figure supplement 3.** Quantitative assessment of the total GAG content in mesenchymal sword of Damocles bone morphogenetic type-2 (MSOD-B) in vitro differentiated constructs across three different batch productions.

**Figure supplement 4.** Quantitative assessment of the total GAG content in mesenchymal sword of Damocles (MSOD)-BΔV1 in vitro differentiated constructs across three different batch productions.

**Figure supplement 5.** Quantitative assessment of the total GAG content in mesenchymal sword of Damocles (MSOD)-BΔR1 in vitro differentiated constructs across three different batch productions.

**Figure supplement 6.** Quantitative assessment of cartilage tissue volume in mesenchymal sword of Damocles bone morphogenetic type-2 (MSOD-B), MSOD-BΔV1, and MSOD-BΔR1 tissues.

remodeling after 6 weeks in vivo. In sharp contrast, the MSOD-BΔR1 only displayed a partial maturation with limited presence of bone and bone marrow.

Altogether, this indicates an incomplete remodeling of RUNX2-edited samples with significantly delayed cortical and trabecular structure formation. This correlates with the impaired hypertrophic phenotype in the corresponding in vitro samples.

We next investigated a potential batch-to-batch variability in the generation of engineered cartilage tissues using the various MSOD-B lines. For each modified lines, independent batches were generated and the amount of glycosaminoglycans was assessed in resulting lyophilized tissues. This revealed a rather consistent generation of cartilage across batches, with no statistical differences across groups (*Figure 4—figure supplements 3 and 4* and *Figure 4—figure supplement 5*). We further conducted a quantitative assessment of pellet volume variability across tissues (*Figure 4—figure supplement 6*), showing minimal differences between the groups, indicating a limited size variation across the samples. Taken together, this points at a low variability across batches of cartilage grafts displaying comparable volume and GAG content.

## RUNX2 knockout in cartilage tissues leads to better cartilage regeneration in a rat osteochondral defect

In order to assess the performance of CRISPR-Cas9-edited eECMs in a relevant skeletal regenerative context, a proof-of-concept study in an immunocompetent rat osteochondral defect was performed. In addition to the lyophilization process, the MSOD-B and MSOD-BΔR1 tissues were also decellularized following a preestablished protocol (*Elder et al., 2009*) in order to reduce inflammation resulting from the rat immune system. We performed a quantitative assessment of the total GAG content in decellularized MSOD-B and MSOD-BΔR1 constructs, showing partial preservation of GAG in the two groups compared to their living counterparts (*Figure 5—figure supplement 1*). We further assessed the total DNA content in MSOD-B and MSOD-BΔR1 constructs before and after decellularization as a measure of decellularization efficiency (*Figure 5—figure supplement 2*). Post-decellularization, the DNA content was significantly reduced to around 540 ng/pellet for Runx2-modified constructs and 280 ng/pellet for MSOD-B grafts (*Figure 5—figure supplement 2*), accounting for a reduction of 98.1% and 97.8% in DNA content respectively indicating an efficient decellularization process across samples. The tissues were subsequently placed in the subchondral defect in the rat distal femur. The defect consisted of a drill hole of 1 mm diameter and 2 mm depth (*Figure 5A*, *Figure 5—figure supplement 3*). Positive controls consisted of untreated healthy rats.

Following explantation, histological stainings and micro-tomography were performed on all sample groups. Hematoxylin and eosin (H&E) and Masson's trichrome indicated de novo bone formation in the defect area for both the MSOD-B and MSOD-BΔR1 groups (*Figure 5B*, *Figure 5—figure supplement 4*). The presence of fibrotic tissue as well as a reduced marrow compartment suggested an incomplete remodeling at that time point (*Figure 5B*). Upon tartrate-resistant acid phosphatase staining, we observed elevated osteoclastic activity in MSOD-B samples compared to MSOD-BΔR1, potentially indicating a reduced rate of active bone formation in MSOD-BΔR1 (*Figure 5C*). The micro-CT quantification revealed a 30% repair of the damaged bone in the defect for both MSOD-B and MSOD-BΔR1 groups (*Figure 5D*, *Figure 5—figure supplement 5*). Of interest, a significantly higher trabecular separation (Tb.Sp) was observed in MSOD-B and MSOD-BΔR1, confirming an ongoing bone remodeling process (*Figure 5E*). Using the ImageJ software, we quantified the trabecular thickness (Tb.Th) in MSOD-B and MSOD-BΔR1 constructs. The analysis revealed no significant differences in Tb.Th between the groups, indicating that trabecular architecture remained consistent across the samples (*Figure 5—figure supplement 6*).

While no statistical differences in bone formation could be identified between the two groups, the MSOD-BΔR1 tissue led to a detectable regeneration of the cartilage area, as revealed by Safranin O staining (*Figure 5F*) in the chondral zone (*Figure 5F, a, b, and c* magnification). Interestingly, the MSOD-BΔR1 group exhibited a higher remnants of GAGs in the subchondral area and better integration (*Figure 5F, d, e, and f* magnification) to the host tissue. Quantification of cartilage tissue within condyle surface area confirmed a poor-to-no repair in the MSOD-B group (1.05%, *Figure 5G*). In sharp contrast, the MSOD-BΔR1 implanted eECMs initiated a chondral regeneration reaching approximately 20.67% of the total healthy cartilage area. The integration and regeneration potential of engineered constructs was further evaluated by performing a semiquantitative histological assessment following

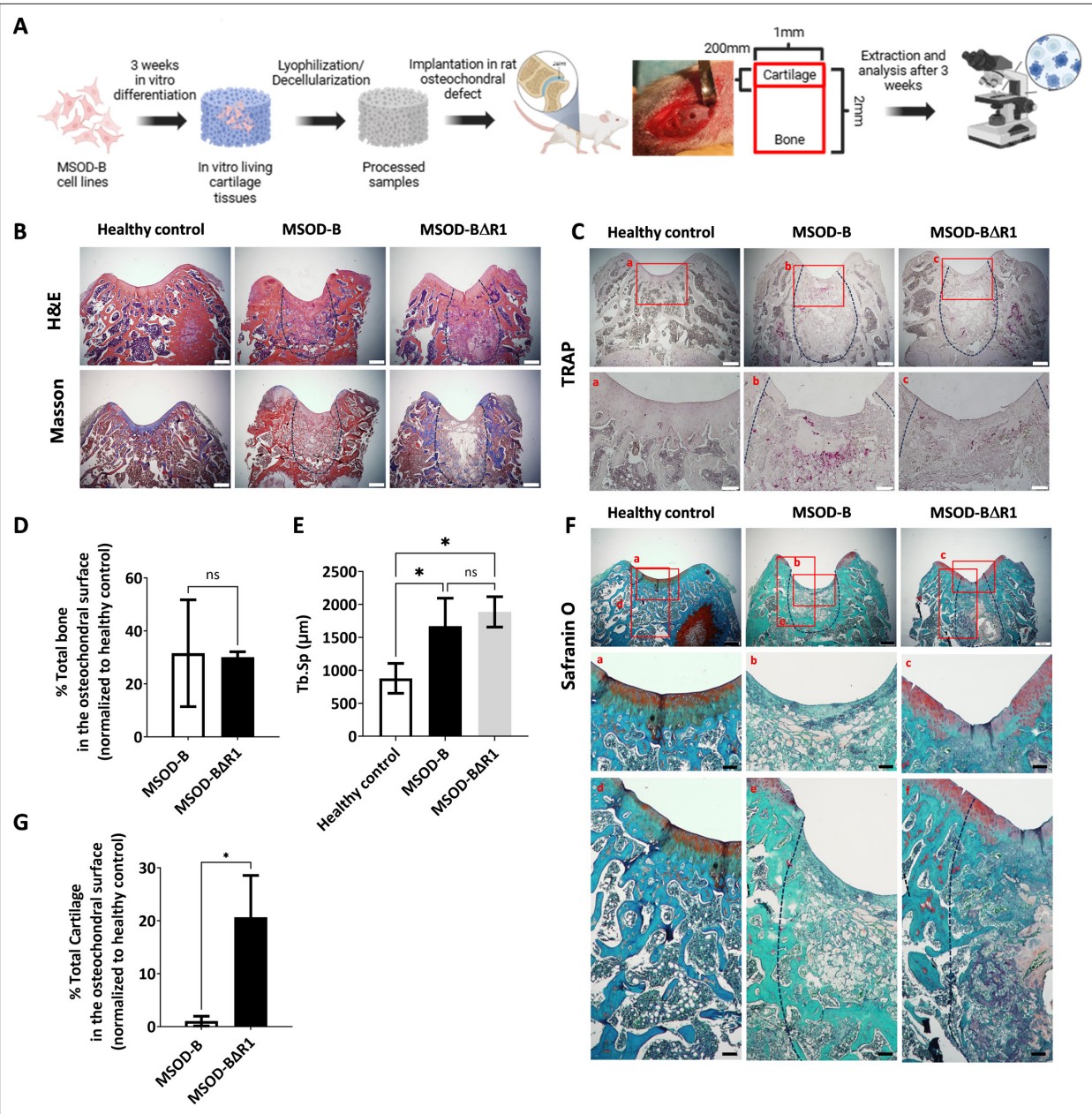

**Figure 5.** Runt-related transcription factor 2 (RUNX2) knockout in cartilage tissues leads to better cartilage regeneration and maintenance in osteochondral defect in rats. (**A**) Experimental scheme for the regenerative potential assessment of mesenchymal sword of Damocles bone morphogenetic type-2 (MSOD-B) and MSOD-BΔR1 cartilage tissues in a rat osteochondral defect. (**B**) Histological analysis of the osteochondral defects for each group using hematoxylin and eosin (H&E) and Masson's trichrome stains, at 3 weeks postimplantation. The dash-line marks the defect area (scale bars=500 μm). (**C**) Histological analysis of the osteochondral defects using tartrate-resistant acid phosphatase (TRAP) staining reporting osteoclastic activity, at 3 weeks postimplantation. The dash-line marks the defect area (scale bars=500 μm and 100 μm for magnified areas). (**D**) Microtomography-based quantification of the sample's total bone/mineralized volume normalized to the healthy control in percentage. Unpaired t-test, n=3 biological replicates, n.s.=not significant. (**E**) ImageJ-based quantification of trabecular separation (Tb.Sp). One-way ANOVA test, n=3 biological replicates, *p<0.05. (**F**) Histological analysis of the osteochondral defects using Safranin O staining. After 3 weeks, a higher regeneration of the surface cartilage is evident in the MSOD-BΔR1 group (**a,b,c**) (scale bars=500 μm and 100 μm for magnified areas). The magnified regions of the subchondral area (**d,e,f**) show higher cartilage remnants and integration in the MSOD-BΔR1 group. (**G**) Quantitative analysis of cartilage regeneration in the osteochondral surface as compared to the healthy control (100%). Unpaired t-test, n=3 biological replicates, *p<0.05. All error bars in the figures indicate the standard deviation (SD). Panel A was created using BioRender.com.

The online version of this article includes the following figure supplement(s) for figure 5:

*Figure 5 continued on next page*

*Figure 5 continued*

**Figure supplement 1.** Quantitative assessment of the total GAG content in mesenchymal sword of Damocles bone morphogenetic type-2 (MSOD-B) and MSOD-BΔR1 in vitro differentiated constructs before (living) and after (decellularized) the decellularization process (unpaired t-test, n=12 biological replicates, ***p<0.001, n.s.=not significant).

**Figure supplement 2.** DNA quantification of in vitro engineered cartilage grafts before (living) and after (decellularized) the decellularization process (unpaired t-test, n=12 biological replicates, ****p<0.0001).

**Figure supplement 3.** Macroscopic images of osteochondral defect model in rats depicting in of empty defect, defect filled with mesenchymal sword of Damocles bone morphogenetic type-2 (MSOD-B), and defect filled with MSOD-BΔR1, respectively.

**Figure supplement 4.** Macroscopic images of extracted rat knees depicting healthy control without any defect, defect filled with mesenchymal sword of Damocles bone morphogenetic type-2 (MSOD-B), and defect filled with MSOD-BΔR1 respectively and microtomography images of same respective constructs.

**Figure supplement 5.** No significant difference in bone volume/total volume (BV/TV) is observed between mesenchymal sword of Damocles bone morphogenetic type-2 (MSOD-B) and MSOD-BΔR1 samples despite significantly reduced BV/TV ratio in both samples compared to healthy control.

**Figure supplement 6.** ImageJ-based quantification of trabecular thickness (Tb.Th).

a preexisting grading system (*Maehara et al., 2010*; *Supplementary file 1* and *Supplementary file 2*). This approach allows for the systematic evaluation of critical repair tissue parameters, offering a comparative measure of the regenerative efficacy of the engineered constructs against healthy tissue benchmarks. In the assessment of stained femur condyle sections, cellular morphology and matrix staining of the MSOD-BΔR1 group (66% and 50%, respectively) were superior to the MSOD-B one (33% and 16%, respectively). Although both constructs exhibited reduced cartilage thickness and subchondral bone regeneration, MSOD-BΔR1 consistently outperformed the MSOD-B grafts (33.33% vs 8.33% and 25% vs 16.66%). Surface regularity and integration (*Figure 5F*) with adjacent cartilage also revealed better outcomes in MSOD-BΔR1 (41% and 50%) as opposed to MSOD-B (50% and 33%), further indicating an enhanced regenerative potential of the MSOD-BΔR1 constructs.

In conclusion, while both grafts yielded comparable bone regeneration within the osteochondral defects, only the RUNX2-deleted grafts supported a cartilage regeneration.

## Discussion

We here report the possibility to edit the composition and function of eECM by CRISPR/Cas9 engineering of human mesenchymal lines. VEGF knockout led to successful in vitro formation of cartilage with targeted protein depletion. Upon implantation, this impacted the graft vascularization onset but not the endogenous instruction of the endochondral program. In turn, RUNX2 knockout prevented cartilage hypertrophy in vitro, significantly delaying ectopic bone and bone marrow formation in vivo. This strategy was validated in a functional osteochondral defect model, whereby prevention of cartilage hypertrophy by RUNX2 deletion improved cartilage regeneration.

A clear interdependency of angiogenesis and osteogenesis occurs upon bone formation (*Kusumbe et al., 2014*; *Tuckermann and Adams, 2021*). For this reason, VEGF-enrichment has been naturally proposed as a strategy to accelerate or increase bone graft vascularization (*Largo et al., 2020*; *Yang et al., 2012*). Our study demonstrates that VEGF knocked-out cartilage templates retained full osteoinductive potential in a challenging ectopic environment. Those results are in sharp contrast with intramembranous strategies (*Burger et al., 2022*; *Behr et al., 2011*), including our previous work (*Bourgine et al., 2017*), whereby grafts remain insufficient in promoting complete bone remodeling even upon VEGF-enrichment. This suggests that the first stage of host progenitor recruitment by our eECM is not VEGF dependent, although subsequent tissue vascularization and remodeling may be orchestrated in a second step by endogenous cell secretion. Other pro-angiogenic proteins embedded in the matrix may also have compensated for the lack of VEGF, such as BMP-2 (*Pigeot et al., 2021*), known to stimulate endothelial cell proliferation, migration, and differentiation (*Zuo et al., 2016*; *Lowery and de Caestecker, 2010*).

Conversely, RUNX2 deletion was shown to fully prevent cartilage hypertrophy, in line with published mouse models also reporting a lack of endochondral ossification and lethality at birth upon RUNX2 knockout (*Kamekura et al., 2006*; *Komori et al., 1998*). Here, MSOD-BΔR cartilage templates still exhibited osteoinductive capacity, but with cortical and trabecular structures largely reduced as compared to controls. Future work may clarify if this strictly results from the absence of COLX, or if an

additional specific cartilage matrix component (e.g. matrix metalloproteinases) impaired the instruction of endochondral ossification.

The performance in a regenerative context was further evaluated in an immunocompetent rat osteochondral defect model (*Meng et al., 2020*). We defined an early time point of 3 weeks as providing the opportunity to assess the early contribution of the grafts to both the cartilage and bone tissue regeneration. Strikingly, only the MSOD-BΔR could contribute to neo-chondrogenesis. We hypothesize that the absence of hypertrophic features prevented the template degradation and favored its integration to native cartilage. Despite the repair performance being limited to an approximate 20% of native cartilage restoration, this is remarkable in light of the BMP-2 content in our ECM prompting endochondral ossification (*Sekiya et al., 2002*; *Pelttari et al., 2006*; *Mueller and Tuan, 2008*). In fact, preventing the in vivo remodeling of engineered cartilage templates remains challenging in the skeletal regeneration field, with bone marrow MSCs cartilage systematically reaching hypertrophy and subsequent ossification. The molecular mechanisms remain nonetheless elusive, warranting further studies comprising additional time points to decipher the repair dynamic as well as the long-term stability of newly formed tissue. It also prompts a comparison with other recently proposed *off-the-shelf* strategies, based on ECM (*Browe et al., 2022*) or synthetic scaffold materials (*Steele et al., 2022*), in order to comprehend the feasibility to exploit our grafts for stable cartilage/joint repair (*Park et al., 2020*; *Zeng et al., 2022*). Taken together, while the relevance of CRISPR/Cas9 eECMs in cartilage repair remains to be further validated, our study demonstrates the relevance of genetically edited ECMs in regenerative contexts.

CRISPR/Cas9 editing of human mesenchymal cells has previously been explored for gene and cell therapy applications (*Golchin et al., 2020*; *Hazrati et al., 2022*), for tailoring the immunomodulatory and/or differentiation capacity of engineered cells. However, no studies have described the exploitation of editing strategies for the generation of eECMs, whereby modified cells are absent from the final grafting product. Our work illustrates that this can be achieved by direct targeting of secreted factors typically embedded upon ECM deposition, as demonstrated with VEGF. Alternatively, we also propose the knockout of key transcription factors as a strategy for custom eECM generation by impacting their tissue developmental/maturation stages.

A clear implication of our work lies in the possibility to decipher the necessary factors capable of instructing de novo tissue formation. Those findings will be of high relevance for the design of eECMs tailored in composition. Beyond bone repair, our study also bears high relevance in other regenerative contexts. In fact, an exciting opportunity also lies in harnessing CRISPR/Cas9 for editing eECMs and tuned their immunogenicity. Key inflammatory components could be turned down, leading to improved efficacy of repair in line with the immuno-engineering principles (*Chen et al., 2016*; *Xie et al., 2020*).

Importantly, our study describes the exploitation of eECMs in a lyophilized form, thus conferring an off-the-shelf storage solution. While the lyophilization and decellularization process can affect the ECM integrity, the resulting grafting products were demonstrated to retain regenerative properties. Together with the standardization of production conferred by stable cell sources, our concept offers exciting translational opportunities. In fact, instrumental to this work is the use of dedicated hMSC lines. The genetic modification of primary cells is laborious, and their limited lifespan ex vivo challenges their selection, characterization, and exploitation for tissue engineering applications (*Aamodt and Grainger, 2016*; *Siddappa et al., 2007*). Here, the MSOD-B was harnessed as an unlimited cell source with robust differentiation potential. This confers a higher standardization potential, although cell line-derived product can also be subject to batch dependency. Our study reports a limited batch-to-batch variation but a stringent characterization of cell lines capacity may be required upon substantial passage. The editing of eECMs remains tedious in part due to the limited CRISPR/Cas9 efficiency and potential off-target effects, calling for a systematic clonal selection. In addition, while large CRISPR/Cas9 screening can be performed in other stem cell systems (*LaFleur et al., 2019*; *Bock et al., 2022*), the critical cell mass required for effective cartilage formation affects parallelization and leads to limited throughput. Nonetheless, after the identification and characterization process, the resulting cell lines can be banked and used for unlimited tissue manufacturing.

The MSOD-B line remains the only human cell source capable of priming endochondral ossification by engineering living or cell-free grafts. This was demonstrated to be driven by the combined low dose of BMP-2 embedded in the tissue (~40 ng/tissue) together with glycosaminoglycans and other

thousands of identified ECM proteins (*Pigeot et al., 2021*). The BMP-2 amount is thus far below the typical amount used in sECMs approaches for effective osteoinduction, falling in the microgram to milligram range (*Pigeot et al., 2021*). This is one key advantage of eECM graft, exhibiting a biological complexity that so far has not been matched by sECMs. Ideally, the two approaches are complementary as the editing of eECMs could inform on the necessary but sufficient factors driving effective regeneration. Those would in turn be embedded in a sECMs strategy and offer a fast and possibly cost-effective solution.

To conclude, the present work offers a platform for decoding factors involved in tissue regeneration and generating tailored eECM. Here, illustrated in the context of skeletal repair, our study may offer similar opportunities in other regenerative situations.

# Materials and methods

## Key resources table

| Reagent type (species) or resource | Designation | Source or reference | Identifiers | Additional information |
|---|---|---|---|---|
| Cell line (*Homo sapiens*) | MSOD-B | University/Hospital of Basel | Pigeot S, Klein T, Gullotta F, et al. doi:10.1002/adma.202103737 | Parental human mesenchymal stromal/stem cell line used for eECM production |
| Cell line (*Homo sapiens*) | MSOD-BΔV1 | MSOD-B (this paper) | MSOD-BΔV1 | CRISPR/Cas9-edited derivative with VEGF knockout. This cell line was created using our MSOD-B line as base. |
| Cell line (*Homo sapiens*) | MSOD-BΔR1 | MSOD-B (this paper) | MSOD-BΔR1 | CRISPR/Cas9-edited derivative with RUNX2 knockout; exhibits impaired hypertrophy. This cell line was created using our MSOD-B line as base. |
| Transfected construct (*Homo sapiens*) | pU6-(BbsI)_CBh-Cas9-T2A-mCherry | Addgene | Plasmid #64324 | Vector for CRISPR/Cas9 editing; encodes SpCas9 and mCherry reporter |
| Transfected construct (*Homo sapiens*) | VEGF gRNAs (VEGF_1.1, 1.2, 1.3, 2.1, 8.1) | This paper | pU6-(BbsI)_CBh-Cas9-T2A-mCherry-VEGF_1.1, 1.2, 1.3, 2.1, 8.1 | Same plasmid but with guide RNAs targeting all isoforms of VEGF |
| Transfected construct (*Homo sapiens*) | RUNX2 gRNAs (RUNX2_2.1, 5.1, 6.1, 6.2, 8.1) | This paper | pU6-(BbsI)_CBh-Cas9-T2A-mCherry-RUNX2_2.1, 5.1, 6.1, 6.2, 8.1 | Same plasmid but with guide RNAs targeting conserved regions of RUNX2 |
| Antibody | Anti-RUNX2 (Rabbit polyclonal) | Thermo Fisher Scientific | Cat# PA5-82787 | Rabbit polyclonal; used for intracellular flow cytometry (1:100) |
| Antibody | Anti-RUNX2 (D1L7F) (Rabbit monoclonal) | Cell Signaling Technology | Cat# 12556 | Used for western blot analysis (1:1000) |
| Antibody | Anti-Actin (Mouse monoclonal) | BD Biosciences | Cat# 612656 | Loading control for western blotting (1:200) |
| Antibody | Anti-Collagen II (Mouse monoclonal) | Invitrogen | Cat# MA137493 | Detects cartilage matrix in immunofluorescence (1:200) |
| Antibody | Anti-Collagen I (Rabbit monoclonal) | Abcam | Cat# ab138492 | Used for extracellular matrix immunostaining (1:200) |
| Antibody | Anti-Collagen X (Rabbit polyclonal) | abbexa | Cat# abx101469 | Marker of hypertrophic cartilage (assesses tissue maturation) (1:200) |
| Antibody | Anti-CD31 (Mouse monoclonal) | R&D Systems | Cat# 11-0319-42 | Vascular endothelial marker; used in immunofluorescence for vessel detection (1:200) |
| Antibody | Anti-VEGF (Rabbit polyclonal) | Bioss Antibodies | Cat# bs-0279R | Used to detect VEGF deposition within engineered ECM (1:200) |
| Sequence-based reagent | RUNX2 | Thermo Fisher Scientific | Hs00298328_s1 | qPCR primer |
| Sequence-based reagent | COLX | Thermo Fisher Scientific | Hs00166657_m1 | qPCR primer |
| Sequence-based reagent | ALPL | Thermo Fisher Scientific | Hs01029144_m1 | qPCR primer |

*Continued on next page*

*Continued*

| Reagent type (species) or resource | Designation | Source or reference | Identifiers | Additional information |
|---|---|---|---|---|
| Sequence-based reagent | Forward primer | IDT | PCR primer | AACGCTTTGTGCTATTTAAGGC |
| Sequence-based reagent | Reverse primer | IDT | PCR primer | AAGAAAGGAACACAAGCAGAGG |
| Sequence-based reagent | Forward primer | IDT | Sequencing primer | TCCCTGTTTTTCTGCTTTTTCC |
| Sequence-based reagent | Reverse primer | IDT | Sequencing primer | TAACTGGGCGGCATTAAATACC |
| Sequence-based reagent | VEGF_1.1F | IDT | gRNA | caccGCGAGCGCCGAGTCGCCACTG |
| Sequence-based reagent | VEGF_1.2 | IDT | gRNA | caccGGAGGAAGAGTAGCTCGCCG |
| Sequence-based reagent | VEGF_1.3 | IDT | gRNA | caccGCCAAGACAGCAGAAAGTTCA |
| Sequence-based reagent | VEGF_2.1 | IDT | gRNA | caccGCTGCACCCATGGCAGAAGG |
| Sequence-based reagent | VEGF_8.1 | IDT | gRNA | caccGTCCTGCCCGGCTCACCGCCT |
| Sequence-based reagent | RUNX2_2.1 | IDT | gRNA | caccTCGTGGGGCGGCCGCAACCG |
| Sequence-based reagent | RUNX2_5.1 | IDT | gRNA | caccTGCGCCCTAAATCACTGAGG |
| Sequence-based reagent | RUNX2_6.1 | IDT | gRNA | caccGCGCCTAGGCACATCGGTGA |
| Sequence-based reagent | RUNX2_6.2 | IDT | gRNA | caccCTAGGCACATCGGTGATGGC |
| Sequence-based reagent | RUNX2_8.1 | IDT | gRNA | caccCATACCGAGGGACATGCCTG |

## Cell expansion

MSOD-B cells and their modified progeny were cultured in a humidified incubator at 37°C and 5% $CO_2$ using complete medium consisting of α-minimum essential medium (αMEM) supplemented with 10% fetal bovine serum, 1% HEPES, 1% sodium pyruvate, 1% penicillin-streptomycin-glutamine solution, and 5 ng/mL of fibroblast growth factor-2 (all from Gibco). Cells were seeded at a density of 3200 cells/cm$^2$ until they reached 90% confluency. The medium was replaced twice a week.

## Chondrogenic and osteogenic differentiation

MSOD-B cells and their modified progeny were harvested from culture flasks by adding Trypsin-EDTA (0.25%) (Gibco) and subsequently seeded on cylindrical Collagen Type I scaffold (Avitene Ultrafoam Collagen Sponge, Davol) of 6 mm in diameter and 3 mm in thickness at a density of 2×10$^6$ cells per scaffold in 12-well plates coated with 1% agarose (Sigma) for chondrogenic differentiation and in a 12-well plate for osteogenic differentiation. Tissue constructs were cultured for 3 weeks in chondrogenic medium (DMEM supplemented with 1% penicillin-streptomycin-glutamine, 1% HEPES [1 M], 1% sodium pyruvate [100 mM], 1% ITS [100×] [Insulin, Transferrin, Selenium] [all from Gibco], 0,47 mg/mL linoleic acid [Sigma], 0.12% bovine serum albumin [25 mg/mL] [BSA] [Sigma], 0.1 mM ascorbic acid [Sigma], 10$^{-7}$ M dexamethasone [Sigma], and 10 ng/mL TGF-β3 [Novartis]). The cells in the 12-well plates were supplemented for 3 weeks with osteogenic (or expansion medium, control) differentiation medium (αMEM with 10% fetal bovine serum, 1% HEPES [1 M], 1% sodium pyruvate [100×10$^{-3}$ M] and 1% penicillin-streptomycin-glutamine solution [100×],

supplemented with 0.01 M β-dexamethasone and 0.1 M ascorbic acid). Media were replaced twice a week.

## Lyophilization

After 3 weeks of chondrogenic differentiation, the tissue constructs were rinsed twice with phosphate-buffered saline (PBS) 7.2 (without calcium/magnesium Gibco), snap-frozen in liquid nitrogen for 5 min and then lyophilized using a freeze dryer (Labconco) (–80°C and 0.05 mbar) overnight. Thereafter, the lyophilized tissue constructs were stored at 4°C.

## Decellularization

After lyophilization, the tissue constructs were treated with a solution containing 1% SDS (Sigma-Aldrich) and DNase I (Sigma-Aldrich) to remove cellular material as according to *Elder et al., 2009*. The constructs were then thoroughly rinsed with PBS (7.2, without calcium/magnesium; Gibco) to eliminate residual chemicals. Following the washing step, the scaffolds were snap-frozen in liquid nitrogen for 5 min and re-lyophilized using a freeze dryer (Labconco) at –80°C and 0.05 mbar overnight. The resulting decellularized tissues were stored at 4°C until experimental use.

## Transfection of MSOD-B cells with gRNAs

MSOD-B cells were seeded at a density of 400,000 cells per well in 12-well plates to reach a minimum of 80% confluency the following day. The medium was replaced before transfection. The transfection was performed with a ratio of 2 µL Lipofectamine to 1 µg DNA. A DNA mix was prepared for each of the five gRNAs and the two plasmid controls. DNA mixtures were composed of 1 µg of plasmid, 2 µL of P3000 reagent, and 50 µL of OptiMEM (Thermo Fisher). A Lipofectamine cocktail was prepared for all DNA mixtures consisting of 24 µL of Lipofectamine 3000 in 400 µL of OptiMEM. Lipofectamine and the DNA mixtures were added at a 1:1 ratio. After 48 hr, cells were analyzed and mCherry-positive clones were FACS-sorted as single cells in 96-well plates using ARIAIII (BD Biosciences). Successfully expanded clones were then further characterized.

## ELISA

VEGF protein content was measured in supernatant collected from cells seeded at 570 cells/cm$^2$ in T175 flask and cultured for 3 days. Content from engineered cartilage tissues was assessed following digestion in RIPA buffer. The Quantikine ELISA kit for Human VEGF-A Immunoassay from the R&D Systems was used according to the manufacturer's instructions to determine protein concentration.

## Intracellular flow cytometry

MSOD-B and MSOD-B ΔRUNX2 cells were trypsinized, fixed, and permeabilized using the Fixation/Permeabilization Kit (BD Biosciences). Following a blocking step in 10% normal donkey serum (Sigma), cells were stained with a primary antibody against RUNX2 (Rabbit anti-human, Thermo Fisher PA5-82787) for 1 hr at room temperature. After primary antibody incubation, cells were washed and stained with an allophycocyanin-labeled secondary antibody (Donkey anti-rabbit IgG DyLight 649, BioLegend 406406) for 30–45 min at room temperature in 2% normal donkey serum. The control samples consisted of unstained cells (negative control) and cells incubated only with the secondary antibody (secondary Ab control). Data were recorded on a BD LSRFortessa Cell Analyzer (BD Biosciences). FCS files were analyzed using the FlowJo software (FlowJo LLC, 10.5.3, BD Biosciences).

## Western blotting

MSOD-B and MSOD-B ΔRUNX2 cells were trypsinized and washed with ice-cold PBS twice and lysed on ice for 10 min in RIPA buffer (#10017003, Thermo Fisher Scientific) supplemented with 1× proteinase and phosphatase inhibitor cocktail (#78440, Thermo Fisher Scientific). The lysates were centrifuged at 16,000×*g* for 15 min at 4°C, and the supernatants were collected. Sample buffer (Laemmli buffer, #161-0737, Bio-Rad) supplemented with 5% 2-mercaptoethanol, 1× proteinase, and phosphatase inhibitor cocktail (#78440, Thermo Fisher Scientific) was added to the supernatant at 1:1 ratio. Samples were boiled at 95°C for 5 min and stored at −80°C or kept on ice until gel loading. Proteins were separated using Bolt gels according to the manufacturer's protocol (#NW04122, #B0002, Thermo Fisher Scientific). iBlot2 system was used to transfer the proteins on polyvinylidene

fluoride (PVDF) membrane membrane according to the manufacturer's protocol (#IB24001, Thermo Fisher Scientific). PVDF membrane was washed once in 1× PBST buffer (#28352, Thermo Fisher Scientific) and blocked in 2% blocking solution (#10156414, Thermo Fisher Scientific) for 1 hr at room temperature. Membranes were incubated overnight at +4°C with primary antibodies at recommended concentrations in 1% blocking solution. Membranes were washed three times (5 min for each wash) with 1× PBST buffer, and secondary HRP-conjugated antibodies in 1% blocking solution were added to the membranes at 1:5000 concentration for 1 hr incubation at room temperature. Membranes were washed three times with 1× PBST and proteins were detected by chemiluminescence according to the manufacturer's protocol (#RPN2232, Thermo Fisher Scientific). The following antibodies were used. Primary antibodies: RunX- (D1L7F-12556) from Cell Signaling Technology and Actin (612656) is from Becton Dickinson. The secondary antibodies were anti-Mouse (GENA931) from Sigma-Aldrich, anti-Rabbit (NA9340V) from Thermo Fisher Scientific.

## Biochemical analysis

Lyophilized tissue constructs were digested by overnight incubation in 0.5 mL of Proteinase K solution (1 mg/mL Proteinase K, Sigma; 10 µg/mL pepstatin A, Sigma; 1 mM EDTA, Sigma; 100 mM Iodoacetamide; 50 mM Tris) at pH 7.6 and 56°C. The GAG content of digested samples was analyzed using Glycosaminoglycan Assay Blyscan kit (Biocolor) following the manufacturer's instruction. DNA residues were quantified using the CyQuant NF Cell Proliferation Assay Kit (Thermo Fisher, USA) following the manufacturer's instructions, with an excitation wavelength of 485 nm and an emission wavelength of 535 nm.

## Mice

FoxN1 KO BALB/C (nude mice) 6–8 weeks of age were obtained from Charles River Laboratories. All mouse experiments and animal care were performed in accordance with the Lund University Animal Ethical Committee guidelines (ethical permit #M15485-18). Mice were housed at a 12 hr light cycle in individually ventilated cages at a positive air pressure and constant temperature. Mice were fed with autoclaved diet and water ad libitum. During the implantation procedure, anesthesia was performed with 2–3% isoflurane (Attane). The mice were kept on a heating pad during the procedure to avoid the heat loss.

## Micro-CT scanning

In vivo samples were explanted and fixed overnight with 4% formaldehyde before ex vivo micro-CT analysis using a U-CT system (MILABS, Netherland) equipped with a tungsten X-ray source at 50 kV and 0.21 mA. Volumes were reconstituted at 10 µm isotropic voxel size. For total volume (TV) analysis, each sample was assessed with Blender (v2.82a, Netherland). Briefly, a mesh was created surrounding the 3D reconstruction of each sample and the volume occupied was then quantified. For bone volume (BV) analysis, the highly mineralized tissue volume was quantified using Seg3D (v2.2.1, NIH, NCRR, Science Computing and Imaging Institute [SCI]).

## Sample preparation for histological analysis

In vitro samples were directly fixed prior to sectioning. In vivo samples were fixed and subsequently decalcified with a 10% EDTA solution, pH=8, at 4°C for 2 weeks prior to tissue embedding.

Paraffin embedding was performed on samples fixed in 4% formalin (Solveca AB). Tissues were dehydrated by immersion in consecutive solutions of 35%, 70%, 95%, and 99.5% graded ethanol solution (Solveco AB). Immersion in a 99.5% ethanol/xylene solution (1:1, Fisher Scientific) was then performed for 10 min followed by two rinses in xylene (Fisher Scientific) for 20 min. Subsequently, tissues were embedded in paraffin at 56°C overnight, before sectioning with a microtome (Microm HM 355 Rotary Microtome) in 5–10 µm sections. The sections were then dried overnight at 37°C before staining. Prior to staining, sections were deparaffinized by two washes in xylene for 7 min and once in 99.5% ethanol/xylene solution (1:1) for 3 min. Afterward, sections were hydrated twice in consecutive solutions of 99.5%, 95%, 70%, and 35% ethanol, for 7 min each.

Agarose embedding was performed on samples fixed overnight with 4% paraformaldehyde (Thermo Scientific), using 4% low-melting agarose (Sigma). Sections of 100 µm thickness were obtained using a 7000smz vibratome (Campden) with stainless steel or ceramic blades.

## Safranin O staining

Paraffin-embedded sections were stained using Mayer's hematoxylin solution (Sigma-Aldrich) for 10 min. Samples were then placed under running distilled water to remove superfluous staining from the sections. Subsequently, sections were stained with 0.01% fast green solution (Fisher Scientific) for 5 min and rinsed with 1% acetic acid solution (glacial, Fisher Scientific) for 15 s. After that, the slides were stained with 0.1% Safranin O (Fisher Scientific) solution for 5 min. Dehydration and clearing were performed by immersion in 95%, 99.5% ethanol, 99.5% ethanol/xylene solution (1:1), and xylene twice successively for 2 min. Finally, the stained slides were mounted with glass slides using PERTEX mounting medium (PERTEX, HistoLab).

## Masson's trichrome staining

Masson's trichrome staining was performed using the trichrome staining kit (Sigma-Aldrich Sweden AB) according to the manufacturer's guidelines. Briefly, tissue sections were deparaffinized and immersed in cold running deionized water for 3 min. Then, the sections were kept in Bouin's solution at room temperature overnight or at 56°C for 15 min. The slides were washed by running tap water and stained using working Weigert's iron hematoxylin solution (Sigma) for 5 min for nuclei detection (in black). After washing the slides, the cytoplasm was stained in red with Biebrich Scarlet-Acid fuchsin for 5 min followed by clearing the slides by immersion in working phosphotungstic/phosphomolybdic acid solution for 5 min. Collagen stained blue by immersion in aniline blue solution for 5 min followed by clearing in 1% acetic acid (glacial, Fisher Scientific) solution diluted in distilled water (glacial, Fisher Scientific) for 2 min and washing with running deionized water. Finally, the sections were dehydrated in graded ethanol solutions (95% once, 100% twice) for 2 min each before washing with xylene twice for 2 min and mounted with PERTEX mounting medium.

## H&E staining

Paraffin-embedded sections were stained using Mayer's hematoxylin solution (Sigma-Aldrich) for 10 min. Samples were then placed under running distilled water to remove superfluous staining from the sections. The slides were immersed precisely in 1× PBS for 1 min to intensify the blue nuclei staining while preserving tissue integrity. Following this, a thorough rinse with three changes of distilled water effectively removed any residual PBS, preparing the sections for the subsequent counterstaining step. Subsequently, Alcoholic-Eosin (Sigma-Aldrich) was applied for 1 min as the counterstain without any rinsing post-application, ensuring optimal interaction of the staining solution with the tissues. Dehydration and clearing were performed by immersion in 95%, 99.5% ethanol, 99.5% ethanol/xylene solution (1:1), and xylene twice successively for 2 min. Finally, the stained slides were mounted with glass coverslips using PERTEX mounting medium (PERTEX, HistoLab).

## Quantitative PCR

Total RNA isolation was conducted from in vitro engineered tissues utilizing the Quick-RNA extraction kit (Zymo Research, R1055) following the manufacturer's protocol. Subsequently, cDNA extraction was carried out using cDNA synthesis kit (Invitrogen 11917020). The quantitative real-time PCR was performed, utilizing the assay on demand from Applied Biosystems to assess the expression of specific genes. These genes include glyceraldehyde 3-phosphate dehydrogenase (GAPDH, HS02786624_G1), runt-related transcription factor 2 (Runx2, Hs00231692_m1), alkaline phosphatase (ALP, Hs01029144_m1), and collagen type X (ColX, Hs00166657_m1).

## Rat osteochondral defect model

Ten- to 12-week-old male Sprague-Dawley rats (n=11) were purchased from JANVIER LABS (France). After 7 days of acclimation, the rats were anesthetized using 3% isoflurane. Then, the animals were placed on a 37°C warm heating pad in prone position. Once anesthetized, isoflurane was lowered to 2– 2.5%, and buprenorphine (Temgesic, 30 µg/kg; Indivior Europe Ltd., Dublin, Ireland) was injected subcutaneously for analgesia. The right hind limb of the animal was shaved and carefully disinfected, and an incision was made along the skin and soft tissue to expose the right knee. After cleaning the knee laterally from soft tissue, patella was displaced laterally to expose the distal femur. An articular cartilage defect with 2 mm depth and 1 mm diameter was created at the trochlear groove of femur. The defect was then push-fitted with the grafts, and not secured with any sutures. The wound was

closed in a layered fashion using resorbable sutures (Vicryl 45-0, Ethicon, Somerville, USA) by closing the joint capsule, followed by the muscle tissue (continuous interlaced suture), followed by closing the skin (Donati suture). Animals started load bearing immediately after surgery. After 3 weeks (n=11) animals were anesthetized by isoflurane inhalation (3%), followed by $CO_2$ asphyxiation, and right-femur were harvested prior to subsequent micro-CT and histological characterization.

## Histology grading method

Histological sections from rat femur condyles were stained with Safranin O, Masson's trichrome, and H&E staining. For each sample, sections were compiled at different depth (top, middle, and bottom) defined to encompass the full tissue characteristics. A semiquantitative analysis of the repaired tissue was performed by utilizing a customized adaptation of the histological grading system initially outlined by *Maehara et al., 2010*. This modified scale encompasses six distinct parameters: cellular morphology, matrix staining, surface regularity, cartilage thickness, regenerated subchondral bone formation, and integration with neighboring cartilage. Each parameter was graded on a numerical scale ranging from 0 to 4 points by independent expert observers, where a maximum total score of 16 indicates the presence of tissue displaying characteristics akin to entirely healthy and normal tissue (*Supplementary file 2*).

## Cartilage quantification method

An evaluation of three distinct sites within the condyle defect was conducted to ascertain the proportion of cartilage in each knee. Utilizing the measured femoral condyle thickness (200 µm) and defect length (1 mm), a standardized region of interest (ROI) was defined using ImageJ. The percentage of cartilage-positive regions within each defect was calculated in relation to the total ROI size, offering a quantitative assessment of cartilage regeneration. This identical approach was applied to evaluate the percentage of cartilage in the healthy control samples.

## Tb.Th and Tb.Sp

Image analysis was performed by ImageJ. To evaluate the defect space, a rectangular interface region ($2 \times 1$ mm$^2$) was defined as the ROI. Tb.Th and Tb.Sp were calculated inside the ROI using BoneJ plugin.

## CAM assay

Fertilized eggs from Lohmann Brown chicken were commercially purchased and incubated 6 days prior (day –6) in a BINDER incubator at 37.5°C with constant humidity. A small window in the shell was opened 3 days prior (day –3) to the start of the experiment under aseptic conditions. The window was resealed with adhesive tape and eggs were returned to the incubator. On day 0, MSOD-B and MSOD-BΔV1 in vitro differentiated and lyophilized samples were placed on top of the CAM. Eggs were resealed and returned to the incubator. Pictures were taken with a bright-field microscope (LEICA S9i) on day 4 and analyzed for vascular density.

## Vascular density quantification

To assess vascular density, the Quantitative Vascular Analysis Tool (Q-VAT) (*Callewaert et al., 2023*) was utilized. The images were acquired using a digital microscope. These images were then segmented into tiles for detailed analysis. The binary vascular masks were generated using the Q-VAT's automated capabilities, to distinguish vascular structures within the tissue. The tool then facilitated the separation of vessel measurements based on their diameters, allowing for a differentiated quantification of macro- and microvasculature. These binary masks were then processed to calculate the vascular density, which refers to the proportion of the tissue area occupied by vessels. This quantitative metric was evaluated for each tile, and the mean vascular density was computed across all tiles for each tissue sample. As part of the analysis, we applied Q-VAT to double- or triple-stained slides to quantify the overlapping percentage of vessels, comparing various time points to observe changes over the course of the experiment.

## Immunofluorescence

Agarose-embedded sections were treated with 0.5% Triton X-100 (Sigma) in PBS supplemented with 20% donkey serum (Jackson ImmunoResearch) for 1 hr at room temperature for blocking and

permeabilization. After blocking/permeabilization, sections were stained with primary antibodies: mouse anti-Collagen II (Invitrogen, MA137493), Anti-Collagen 1 (Abcam, ab138492), VEGF polyclonal antibody (Bioss Antibodies, bs-0279R), rabbit anti-Collagen Type X (ColX) (abbexa, abx101469), Goat anti mouse/rat CD31 (R&D Systems, Cat# 11-0319-42), VEGF at 4°C overnight. Sections were washed three times using cold PBS with 0.1% Triton X-100, 20 min each, followed by staining with secondary antibodies: CF568 donkey anti-rabbit (Biotium, 20098), CF567 anti-goat (Invitrogen, A11057), CF633 donkey anti-mouse (Sigma-Aldrich, SAB4600131 in 2% donkey serum for 3 hr at room temperature). Sections were washed three times using cold PBS with 0.1% Triton X-100, 20 min each once more.

All sections were mounted using the Vectashield antifade mounting medium containing DAPI (Vector Laboratories, H1200). An LSM780 confocal microscope (Zeiss) with a ×10/×20 objective was used to capture images and image series of whole tissue sections (z-step=2 µm). The IMARIS 9.5 software package (Oxford Instruments) was used to analyze the data.

## Acknowledgements

We express our gratitude to the following facilities and individuals for their invaluable support and contributions to this study: FACS Facility (Anna Hammarberg), Cell and Gene Therapy Core Facility (Pia Johansson) and Imaging Facility at Lund Stem Cell Centre, as well as the Lund University Bioimaging Centre, for providing access to the X-ray CT system. This research study was made possible by financial support through the Knut and Alice Wallenberg grant, the Medical Faculty at Lund University, the Swedish Research Council (Starting grant to PEB), the European Research Council (Starting grant hOssicle #948588 to PEB), IngaBritt och Arne Lundbergs Forskningsstiftelse (grant to PEB), and Region Skåne. Figures were generated using BioRender. Special thanks to Ani Grigoryan for her expertise and valuable insights regarding FACS experiments. Christian Hansen deserves our appreciation for providing the mCherry plasmid and offering valuable guidance and knowledge related to CRISPR-Cas9.

## Additional information

### Funding

| Funder | Grant reference number | Author |
| --- | --- | --- |
| Vetenskapsrådet | 2019-01864_3 | Paul E Bourgine |
| European Research Council | 948588 | Paul E Bourgine |
| IngaBritt och Arne Lundbergs Forskningsstiftelse | | Paul E Bourgine |
| Knut and Alice Wallenberg Foundation | | Paul E Bourgine |
| Lund University Medical Faculty Foundation | | Paul E Bourgine |
| Region Skåne | | Paul E Bourgine |

The funders had no role in study design, data collection and interpretation, or the decision to submit the work for publication.

### Author contributions

Sujeethkumar Prithiviraj, Conceptualization, Data curation, Formal analysis, Validation, Methodology, Writing – original draft; Alejandro Garcia Garcia, Bai Yiguang, Data curation, Formal analysis, Validation, Methodology, Writing – review and editing; Karin Linderfalk, Sonia Ferveur, Ludvig Nilsén Falck, Steven J Dupard, Data curation, Formal analysis, Writing – review and editing; Agatheeswaran Subramaniam, Formal analysis, Supervision, Writing – review and editing; Sofie Mohlin, Supervision, Methodology, Writing – review and editing; David Hidalgo Gil, Data curation, Formal analysis, Methodology, Writing – review and editing; Dimitra Zacharaki, Data curation, Validation;

Deepak Bushan Raina, Data curation, Formal analysis, Supervision, Validation, Methodology, Writing – review and editing; Paul E Bourgine, Conceptualization, Formal analysis, Supervision, Funding acquisition, Investigation, Writing – original draft, Project administration, Writing – review and editing

### Author ORCIDs
Sujeethkumar Prithiviraj ⓘ http://orcid.org/0000-0001-8314-4234
Sofie Mohlin ⓘ https://orcid.org/0000-0002-2458-3963
Paul E Bourgine ⓘ https://orcid.org/0000-0002-7639-6844

### Ethics
This study was performed in strict accordance with the recommendations in the Guide for the Care and Use of Laboratory Animals of the Swedish Board of Agriculture. All of the animals were handled according to approved institutional animal care and the Malmo - Lund's animal welfare organization. The protocol was approved by the Malmo - Lund's animal welfare organization committee on the Ethics of Animal Experiments (Permit Number: 15485-18 and 11518-20). All surgery was performed under anesthesia, and every effort was made to minimize suffering.

Reviewer #1 (Public review): https://doi.org/10.7554/eLife.96941.3.sa1
Reviewer #3 (Public review): https://doi.org/10.7554/eLife.96941.3.sa2
Author response https://doi.org/10.7554/eLife.96941.3.sa3

## Additional files

### Supplementary files
Supplementary file 1. Scoring rate for individual categories. Semiquantitative analysis utilizing a modified histological grading system, including parameters such as cellular morphology, matrix staining, surface regularity, thickness of cartilage, subchondral bone formation, and integration of adjacent cartilage, resulting in a comprehensive assessment of tissue regeneration and integration

Supplementary file 2. Histological grading system. The histological graded system is modified and adapted from previously established model as mentioned in Materials and methods section. This modified scale incorporates six distinct parameters to comprehensively evaluate the quality of tissue repair.

MDAR checklist

Source data 1. Compiled source data for all figures.

### Data availability
All data generated or analysed during this study are included in the manuscript and supporting files have been provided for all Figures.

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
