## [Editor Report · eLife Assessment]

The study presents a potentially **valuable** approach to genetically modify cells to produce extracellular matrices with altered compositions, termed cell-laid, engineered extracellular matrices (eECM). The evidence supporting the authors' conclusions regarding the utility of eECM for endogenous repair is **solid**, although there are some disagreements on the chondrogenicity of lyophilized constructs which was viewed as lacking robust evidence for endochondral ossification.

---

## [Referee Report · Reviewer #1 (Public review)]

Summary:

The authors aimed to modify the characteristics of the extracellular matrix (ECM) produced by immortalized mesenchymal stem cells (MSCs) by employing the CRISPR/Cas9 system to knock out specific genes. Initially, they established VEGF-KO cell lines, demonstrating that these cells retained chondrogenic and angiogenic properties. Additionally, lyophilized carriage tissues produced by these cells exhibited retained osteogenic properties.

Subsequently, the authors established RUNX2-KO cell lines, which exhibited reduced COLX expression during chondrogenic differentiation and notably diminished osteogenic properties in vitro. Transplantation of lyophilized carriage tissues produced by RUNX2-KO cell lines into osteochondral defects in rat knee joints resulted in the regeneration of articular cartilage tissues as well as bone tissues, a phenomenon not observed with tissues derived from parental cells. This suggests that gene-edited MSCs represent a valuable cell source for producing ECM with enhanced quality.

Strengths:

The enhanced cartilage regeneration observed with ECM derived from RUNX2-KO cells supports the authors' strategy of creating gene-edited MSCs capable of producing ECM with superior quality. Immortalized cell lines offer a limitless source of off-the-shelf material for tissue regeneration.

Weaknesses:

Most of the data align with anticipated outcomes, offering limited novelty to advance scientific understanding. Methodologically, the chondrogenic differentiation properties of immortalized MSCs appeared deficient, evidenced by Safranin-O staining of 3D tissues and histological findings lacking robust evidence for endochondral differentiation. This presents a critical limitation, particularly as authors propose the implantation of cartilage tissues for in vivo experiments. Instead, the bulk of data stemmed from type I collagen scaffold with factors produced by MSCs stimulated by TGFβ.

In the revised version, the authors presented Safranin-O staining results of pellets prior to lyophilization. The inset of figures showing entire pellets revealed that Safranin-O-positive areas were limited, suggesting that cells in the negative regions had not differentiated into chondrocytes. In Figure 3F, DAPI staining showed devitalized cells in the outer layer but was negative in the central part, indicating the absence of cells in these areas and incomplete differentiation induction.

The rationale for establishing VEGF-KO cell lines remains unclear, and the authors' explanation in the revised manuscript is still equivocal. While they mention that VEGF is a late marker for endochondral ossification, the data in Figures 1D and 1E clearly show that VEGF-KO affects the early phase of endochondral ossification.

Insufficient depth was given to elucidate the disparity in osteogenic properties between those observed in ectopic bone formation and those observed in transplantation into osteochondral defects.

In the ectopic bone formation study, most of the collagenous matrix observed at 2 weeks was resorbed by 6 weeks, with only a small amount contributing to bone formation in MSOD-B cells (Figs. 2I and 4C). This finding does not align with the micro-CT data presented in Figures 2H and 4B. For the micro-CT experiments, it would be more appropriate to use a standard window for bone and present the data accordingly.

While the regeneration of articular cartilage in RUNX2-KO ECM presents intriguing results, the study lacked an exploration into underlying mechanisms, such as histological analyses at earlier time points.

---

## [Referee Report · Reviewer #3 (Public review)]

Summary:

In this study, the authors have started off using an immortalized human cell line and then gene edited it to decrease the levels of VEGF1 (in order to influence vascularization), and the levels of Runx2 (to decrease osteogenesis). They first transplanted these cells with a collagen scaffold. The modified cells showed a decrease in vascularization when VEGF1 was decreased, and suggested an increase in cartilage formation.

In another study, matrix generated by these cells subsequently remodeled into a bone marrow organ. When RUNX2 was decreased, the cells did not mineralize in vitro, and their matrices expressed types I and II collagen but not type X collagen in vitro, in comparison with unedited cells. In vivo, the author claims that remodeling of the matrices into bone was somewhat inhibited. Lastly, they utilized matrices generated by RUNX2-edited cells to regenerate chondro-osteal defects. They suggest that the edited cells regenerated cartilage in comparison with unedited cells.

Strengths:

- The notion that inducing changes in the ECM by genetically editing the cells is a novel one, as it has long been thought that ECM composition influences cell activity.

- If successful, it may be possible to make off the shelf ECMS to carry out different types of tissue repair.

Weaknesses:

- The authors have not demonstrated robust cartilage formation (quantitation would be useful).

- Measuring total GAG content does not prove the presence of cartilage

- There are numerous overstatements about forming and implanting cartilage.

- Although it is implied, RUNX2 deletion did not improve cartilage formation by the modified cells.

- In the control line, MSOD-B there were variability in the amount of safranin O positive material in various histological panels in the figures.; more quantitation is needed.

- In the in vivo articular defect experiments, an untreated injured joint is needed as a negative control.

- Statements about bone generation are often not reflective of the microCT data presented.

- The discussion over-interprets the results.

---

## [Author Response]

The following is the authors’ response to the current reviews.

**eLife Assessment**
The study presents a potentially valuable approach to genetically modify cells to produce extracellular matrices with altered compositions, termed cell-laid, engineered extracellular matrices (eECM). The evidence supporting the authors' conclusions regarding the utility of eECM for endogenous repair is solid, although there are some disagreements on the chondrogenicity of lyophilized constructs which was viewed as lacking robust evidence for endochondral ossification.

We thank the reviewers for the assessment of our work. We however strongly contest the lack of evidence for chondrogenicity and endochondral ossification. This is robustly demonstrated and a clear strength of our study.

**Public Reviews:**

**Reviewer #1 (Public review):**
Summary:The authors aimed to modify the characteristics of the extracellular matrix (ECM) produced by immortalized mesenchymal stem cells (MSCs) by employing the CRISPR/Cas9 system to knock out specific genes. Initially, they established VEGF-KO cell lines, demonstrating that these cells retained chondrogenic and angiogenic properties. Additionally, lyophilized carriage tissues produced by these cells exhibited retained osteogenic properties.Subsequently, the authors established RUNX2-KO cell lines, which exhibited reduced COLX expression during chondrogenic differentiation and notably diminished osteogenic properties in vitro. Transplantation of lyophilized carriage tissues produced by RUNX2-KO cell lines into osteochondral defects in rat knee joints resulted in the regeneration of articular cartilage tissues as well as bone tissues, a phenomenon not observed with tissues derived from parental cells. This suggests that gene-edited MSCs represent a valuable cell source for producing ECM with enhanced quality.Strengths:The enhanced cartilage regeneration observed with ECM derived from RUNX2-KO cells supports the authors' strategy of creating gene-edited MSCs capable of producing ECM with superior quality. Immortalized cell lines offer a limitless source of off-the-shelf material for tissue regeneration.Weaknesses:Most of the data align with anticipated outcomes, offering limited novelty to advance scientific understanding. Methodologically, the chondrogenic differentiation properties of immortalized MSCs appeared deficient, evidenced by Safranin-O staining of 3D tissues and histological findings lacking robust evidence for endochondral differentiation. This presents a critical limitation, particularly as authors propose the implantation of cartilage tissues for in vivo experiments. Instead, the bulk of data stemmed from type I collagen scaffold with factors produced by MSCs stimulated by TGFβ.

We thank the reviewer for the thorough evaluation. We appreciate the highlighted novelty but overall disagree with key points from the provided assessment. The most important one being non the contested in vitro cartilage and endochondral ossification by engineered ECMs, for which we have provided compelling evidence. Of note, the reviewer points the “osteogenic” properties of our tissues; the wording is incorrect since cells are absent from the final grafts. Here, the term ”osteoinductivity” should be employed, in line with the model of ectopic ossification used to demonstrate de novo bone formation.

In the revised version, the authors presented Safranin-O staining results of pellets prior to lyophilization. The inset of figures showing entire pellets revealed that Safranin-O-positive areas were limited, suggesting that cells in the negative regions had not differentiated into chondrocytes. In Figure 3F, DAPI staining showed devitalized cells in the outer layer but was negative in the central part, indicating the absence of cells in these areas and incomplete differentiation induction.

We strongly disagree with the reviewer on the lack of demonstrated chondrogenicity. We have provided evidence of Safranin-O positivity, GAGs quantification, as well as collagen type 2 and collagen type X stainings (also quantified). Frankly, those are gold standard assays in the field and we do not understand the reviewer point of view. We however agree that our grafts are not entirely composed of cartilage matrix. There are areas where cartilage is absent, in particular in the core of the tissues. This is expected from in vitro engineered cartilage pellets even from primary BM-MSCs donors. By selecting primary donors it is possible to obtain a superior cartilage formation. Our MSOD-B cells remain to-the-best-of-our -knowledge, the only human line capable of in vitro chondrogenesis, even if considered moderate.

We agree with the absence of cells in the core area of our tissues, as correctly pointed out by the reviewer. This has been reported in other studies whereby the lack of media diffusion can lead to necrotic core formation.

The rationale for establishing VEGF-KO cell lines remains unclear, and the authors' explanation in the revised manuscript is still equivocal. While they mention that VEGF is a late marker for endochondral ossification, the data in Figures 1D and 1E clearly show that VEGF-KO affects the early phase of endochondral ossification.

We feel that the rationale for a VEGF-KO is sufficiently conveyed. In our study, VEGF-KO affects GAGs content in the tissue, but not the efficiency of ossification.

Insufficient depth was given to elucidate the disparity in osteogenic properties between those observed in ectopic bone formation and those observed in transplantation into osteochondral defects.

We here agree with the reviewer on the limited depth of our osteochondral assessment. However, this was performed as a proof-of-concept and we clearly conveyed both limitations and need of a follow-up study to demonstrate the repair efficacy of our tissue in such defect context.

In the ectopic bone formation study, most of the collagenous matrix observed at 2 weeks was resorbed by 6 weeks, with only a small amount contributing to bone formation in MSOD-B cells (Figs. 2I and 4C). This finding does not align with the micro-CT data presented in Figures 2H and 4B. For the micro-CT experiments, it would be more appropriate to use a standard window for bone and present the data accordingly.

Stainings report the deposition of collagens and may be misleading as not only indicating frank bone formation. This is the reason why we provided microCT data, offering a quantitative assessment of the full grafts and more reliably evaluating mineralized/bone tissue. We feel that our results matched our conclusions.

While the regeneration of articular cartilage in RUNX2-KO ECM presents intriguing results, the study lacked an exploration into underlying mechanisms, such as histological analyses at earlier time points.

We do agree with the reviewer regarding this limitation. In addition to mechanisms and early timepoints, we are also interested in longer in vivo evaluation. This represents a significant amount of work which is beyond the scope of our present manuscript.

**Reviewer #3 (Public review):**
Summary:In this study, the authors have started off using an immortalized human cell line and then gene edited it to decrease the levels of VEGF1 (in order to influence vascularization), and the levels of Runx2 (to decrease osteogenesis). They first transplanted these cells with a collagen scaffold. The modified cells showed a decrease in vascularization when VEGF1 was decreased, and suggested an increase in cartilage formation.In another study, matrix generated by these cells subsequently remodeled into a bone marrow organ. When RUNX2 was decreased, the cells did not mineralize in vitro, and their matrices expressed types I and II collagen but not type X collagen in vitro, in comparison with unedited cells. In vivo, the author claims that remodeling of the matrices into bone was somewhat inhibited. Lastly, they utilized matrices generated by RUNX2-edited cells to regenerate chondro-osteal defects. They suggest that the edited cells regenerated cartilage in comparison with unedited cells.Strengths:- The notion that inducing changes in the ECM by genetically editing the cells is a novel one, as it has long been thought that ECM composition influences cell activity.- If successful, it may be possible to make off the shelf ECMS to carry out different types of tissue repair.Weaknesses:- The authors have not demonstrated robust cartilage formation (quantitation would be useful).- Measuring total GAG content does not prove the presence of cartilage- There are numerous overstatements about forming and implanting cartilage.- Although it is implied, RUNX2 deletion did not improve cartilage formation by the modified cells.- In the control line, MSOD-B there were variability in the amount of safranin O positive material in various histological panels in the figures.; more quantitation is needed.- In the in vivo articular defect experiments, an untreated injured joint is needed as a negative control.- Statements about bone generation are often not reflective of the microCT data presented.- The discussion over-interprets the results.

We thank the reviewer for the further assessment of our work. We respectfully disagree with most of the provided statements. The chondrogenicity of our graft is robustly demonstrated using multiple readouts, including quantitative ones. Beyond GAGs, we provided clear Safranin-O stainings, as well as collagen type 2 and X indicating presence of hypertrophic cartilage matrix. Those are the gold standards in the field and we thus do not understand the reviewer scepticism. We do agree that our grafts are fully composed of cartilage matrix, with areas (in the core) deprived of cartilage. This does not impact the core findings of our study and its conclusions, and we strongly feel our statements about forming in vitro cartilage fully stand.

We do not claim in the manuscript an increased cartilage formation following RUNX2 deletion. We report in vitro an impaired hypertrophy (collagen type X) and maintenance of collagen type 2 and GAGs content.

We are confident on our data regarding de novo bone formation bi priming endochondral ossification, confirmed both by stainings and microCT. We feel that our claims are well-supported.

The following is the authors’ response to the original reviews.

**Public Reviews:**

**Reviewer #1 (Public Review):**
Summary:The authors aimed to modify the characteristics of the extracellular matrix (ECM) produced by immortalized mesenchymal stem cells (MSCs) by employing the CRISPR/Cas9 system to knock out specific genes. Initially, they established VEGF-KO cell lines, demonstrating that these cells retained chondrogenic and angiogenic properties. Additionally, lyophilized carriage tissues produced by these cells exhibited retained osteogenic properties.Subsequently, the authors established RUNX2-KO cell lines, which exhibited reduced COLX expression during chondrogenic differentiation and notably diminished osteogenic properties in vitro. Transplantation of lyophilized carriage tissues produced by RUNX2-KO cell lines into osteochondral defects in rat knee joints resulted in the regeneration of articular cartilage tissues as well as bone tissues, a phenomenon not observed with tissues derived from parental cells. This suggests that gene-edited MSCs represent a valuable cell source for producing ECM with enhanced quality.Strengths:The enhanced cartilage regeneration observed with ECM derived from RUNX2-KO cells supports the authors' strategy of creating gene-edited MSCs capable of producing ECM with superior quality. Immortalized cell lines offer a limitless source of off-the-shelf material for tissue regeneration.

We thank the reviewer for the interest in our work. We however want to clarify that the present manuscript does not report the generation of ECM with “superior quality”, but rather of modulated composition and thus function.

Weaknesses:Most data align with anticipated outcomes, offering limited novelty to advance scientific understanding. Methodologically, the chondrogenic differentiation properties of immortalized MSCs appeared deficient, evidenced by Safranin-O staining of 3D tissues and histological findings lacking robust evidence for endochondral differentiation. This presents a critical limitation, particularly as authors propose the implantation of cartilage tissues for in vivo experiments. Instead, the bulk of data stemmed from type I collagen scaffold with factors produced by MSCs stimulated by TGFβ.

The chondrogenic differentiation of our MSOD-B line and their capacity of undergoing endochondral ossification has been robustly demonstrated in previous studies (Pigeot et al., Advanced Materials 2021 and Grigoryan et al., Science Translational Medicine 2022). In the present manuscript, we thus compare the chondrogenic capacity of newly established VEGF-KO and RUNX-KO lines to those of MSOD-B cells. We demonstrate by qualitative (Safranin-O staining, Collagen type 2 and Collagen type X immuno-stainings) and quantitative (glycosaminoglycans assay) assays that the generated tissues consist in cartilage grafts of similar quality than the MSOD-B counterpart. Of note, the safranin-O stainings were performed on lyophilized tissues, which can alter the staining quality/intensity. We now provide additional stainings of generated tissues pre-lyophilization. This is implemented in Figure 1D, Figure 3D.

The rationale behind establishing VEGF-KO cell lines remains unclear. What specific outcomes did the authors anticipate from this modification?

VEGF is a known master regulator of angiogenesis and a key mediator of endochondral ossification. It has also been extensively used in bone tissue engineering studies as a supplemented factor – primarily in the form of VEGFα – to increase the vascularization and thus outcome of bone formation of engineered grafts (https://www.nature.com/articles/s42003-020-01606-9, https://www.sciencedirect.com/science/article/pii/S8756328216301752). In our study, it was thus identified as a natural candidate to demonstrate the possibility to generate VEGF-KO cartilage and subsequently assess the functional impact on both the angiogenic and osteogenic potential of resulting cartilage tissue. This is now clarified in the manuscript (page 3, paragraph 4).

Insufficient depth was given to elucidate the disparity in osteogenic properties between those observed in ectopic bone formation and those observed in transplantation into osteochondral defects. While the regeneration of articular cartilage in RUNX2-KO ECM presents intriguing results, the study lacked an exploration into underlying mechanisms, such as histological analyses at earlier time points.

Using RUNX2-KO ECM, we aimed at demonstrating the impact on cartilage remodeling and bone formation. This was performed ectopically but also in the rat osteochondral defect as a regenerative set-up of higher clinical relevance. We agree with the reviewer that additional experimental groups and time-points (not only earlier but also longer ones) would offer a better mechanistic understanding of the ECM contribution to the joint repair. However, as stated in our manuscript this is a proof-of-concept study that successfully demonstrated the influence of the cartilage ECM modification on the in vivo skeletal regeneration. A follow-up study would need to be performed to complement existing evidence and strengthen the relevance of our approach for cartilage repair. This is now further emphasized in the discussion (page 11, paragraph 3).

**Reviewer #2 (Public Review):**
The manuscript submitted by Sujeethkumar et al. describes an alternative approach to skeletal tissue repair using extracellular matrix (ECM) deposited by genetically modified mesenchymal stromal/stem cells. Here, they generate a loss of function mutations in VEGF or RUNX2 in a BMP2overexpressing MSC line and define the differences in the resulting tissue-engineered constructs following seeding onto a type I collagen matrix in vitro, and following lyophilization and subcutaneous and orthotopic implantation into mice and rats. Some strengths of this manuscript are the establishment of a platform by which modifications in cell-derived ECM can be evaluated both in vitro and in vivo, the demonstration that genetic modification of cells results in complexity of in vitro cell-derived ECM that elicits quantifiable results, and the admirable goal to improve endogenous cartilage repair. However, I recommend the authors clarify their conclusions and add more information regarding reproducibility, which was one limitation of primary-cell-derived ECMs.

We thank the reviewer for the positive evaluation of our work.

Overcoming the limitations of native/autologous/allogeneic ECMs such as complete decellularization and reduction of batch-to-batch variability was not specifically addressed in the data provided herein. For the maintenance of ECM organization and complexity following lyophilization, evidence of complete decellularization was not addressed, but could be easily evaluated using polarized light microscopy and quantification of human DNA for example in constructs pre and post-lyophilization.

We appreciate the reviewer comments and acknowledge the lack of information in the first version of our manuscript. In line with our previous study (Pigeot et al., Advanced Materials 2021), the ectopic evaluation of our cartilage pellets was strictly done with lyophilized tissues using immunocompromised animals. Lyophilized tissues are thus considered devitalized, and not decellularized. Instead, the osteochondral defect experiment was performed with decellularized tissues in order to be able to implant the grafts in the rat immuno-competent model. This is now specified consistently throughout the manuscript. The decellularization process is also now incorporated accordingly in the method section (page 14, paragraph 2). We also provide quantifications of GAGs and DNAs from tissue pre- and post-decellularization (Supplementary figure 6A and 6B), described in the result section of the manuscript (page 9, paragraph 1). The decellularization step led to 97-98% of DNA removal.

Importantly, we do not claim full maintenance of ECM integrity following lyophilization nor decellularization. This is now clarified in the discussion (page 12, paragraph 2). However, we report their capacity to instruct skeletal regeneration in multiple contexts despite extensive processing.

It would be ideal to see minimization of batch-to-batch variability using this approach, as mitigation of using a sole cell line is likely not sufficient (considering that the sole cell line-derived Matrigel does exhibit batch-to-batch and manufacturer-to-manufacturer variability). I recommend adding details regarding experimental design and outcomes not initially considered. Inter- and intraexperimental reproducibility was not adequately addressed. The size of in vitro-derived cartilage pellets was not quantified, and it is not clear that more than one independent 'differentiation' was performed from each gene-edited MSC line to generate in vitro replicates and constructs that were implanted in vivo.

We thank the Reviewer for the comment on variability/reproducibility concern. Using a cell line does confer higher robustness but indeed does not grant unlimited consistency of batch production. We now temper our claims in the discussion and mention the need to regularly recharacterize cell lines properties upon passages (page 12, paragraph 2). Using our edited lines, we have generated multiple batches of cartilage grafts for their in vitro characterization or in vivo performance assessment. We have now compiled batch variations of GAG content and pellet volume, provided as Supplementary figure 5. This revealed that batches are indeed not identical (nor each pellets), but the production remains consistent.

The use of descriptive language in describing conclusions may mislead the reader and should be modified accordingly throughout the manuscript. For example, although this reviewer agrees with the comparative statements made by the authors regarding parental and gene-edited MSC lines, non-quantifiable terms such as 'frank' 'superior' (example, line 242) are inappropriate and should rather be discussed in terms of significance. Another example is 'rich-collagenous matrix,' which was not substantiated by uniform immunostaining for type II collagen (line 189).

We thank the Reviewer for the constructive suggestions. We have revised the language accordingly throughout the manuscript.

I have similar recommendations regarding conclusive statements from the rat implantation model, which was appropriately used for the purpose of evaluating the response of native skeletal cells to the different cell-derived ECMs. Interpretations of these results should be described with more accuracy. For example, increased TRAP staining does not indicate reduced active bone formation (line 237). Many would not conclude that GAGs were retained in the RUNX2-KO line graft subchondral region based on the histology. Quantification of % chondral regeneration using histology is not accurate as it is greatly influenced by the location in the defect from which the section was taken. Chondral regeneration is usually semi-quantified from gross observations of the cartilage surface immediately following excision. The statements regarding integration (example line 290) are not founded by histological evidence, which should show high magnification of the periphery of the graft adjacent to the native tissue.

We have revised our language relative to the TRAP staining description (page 9, paragraph 2). We also agree with the reviewer on the semi-quantitative approach of our methodology, which we transparently disclosed both in the main text (page 9, paragraph 3) and method section (page 18, paragraph 2). The sectioning location does influence the analysis, but to prevent this we performed an assessment at different depth (top, middle, bottom for each sample). This is now implemented in our method section (page 18, paragraph 3). On the tissue integration, we now provide higher magnification images of the implant/host tissue area (Figure 5F).

**Reviewer #3 (Public Review):**
Summary:In this study, the authors have started off using an immortalized human cell line and then geneedited it to decrease the levels of VEGF1 (in order to influence vascularization), and the levels of Runx2 (to decrease chondro/osteogenesis). They first transplanted these cells with a collagen scaffold. The modified cells showed a decrease in vascularization when VEGF1 was decreased, and suggested an increase in cartilage formation.In another study, the matrix generated by these cells was subsequently remodeled into a bone marrow organ. When RUNX2 was decreased, the cells did not mineralize in vitro, and their matrices expressed types I and II collagen but not type X collagen in vitro, in comparison with unedited cells. In vivo, the author claims that remodeling of the matrices into bone was somewhat inhibited. Lastly, they utilized matrices generated by RUNX2 edited cells to regenerate chondro-osteal defects. They suggest that the edited cells regenerated cartilage in comparison with unedited cells.Strengths:- The notion that inducing changes in the ECM by genetically editing the cells is a novel one, as it has long been thought that ECM composition influences cell activity.- If successful, it may be possible to make off-the-shelf ECMS to carry out different types of tissue repair.

We thank the Reviewer for the critical evaluation of our work and the highlighted novelty of it.

Weaknesses:- The authors have not generated histologically identifiable cartilage or bone in their transplants of the cells with a type I scaffold.

The chondrogenic differentiation of our MSOD-B line and their capacity of undergoing endochondral ossification has been robustly demonstrated in previous studies (Pigeot et al., Advanced Materials 2021 and Grigoryan et al., Science Translational Medicine 2022). In the present manuscript, we thus compare the chondrogenic capacity of newly established VEGF-KO and RUNX-KO lines to those of MSOD-B. We demonstrate by qualitative (Safranin-O staining, Collagen type 2 and Collagen type X immuno-stainings) and quantitative (glycosaminoglycans assay) assays that the generated tissues consist in cartilage tissue of similar quality than the MSOD-B. Of note, the safranin-O stainings were performed on lyophilized tissues, which can alter the staining quality/intensity. We now provide here additional stainings of generated tissues pre-lyophilization. This is implemented in Figure 1D and Figure 3D.

On the contested formation of bone in vivo by our ECMs grafts, we have provided compelling qualitative evidence via Masson´s Trichrome stainings and quantification of mineralized volume by µCT. Both cortical bone and trabecular structures were identified ectopically. Those are standard evaluation methods in the field, we would be happy to receive additional suggestions by the Reviewer.

- In many cases, they did not generate histologically identifiable cartilage with their cell-free-edited scaffold. They did generate small amounts of bone but this is most likely due to BMPs that were synthesized by the cells and trapped in the matrix.

We now appreciate that the Reviewer agrees on the successful formation of bone induced by our engineered grafts. We however still respectfully disagree with the “small amount of bone” statement since our MSOD-B and MSOD-B VEGF KO cartilage grafts led to the full generation of a mature ectopic bone organ (that is, also composed of extensive marrow). This has been assessed qualitatively and quantitatively.

We agree with the Reviewer on the key role of BMP-2 in the remodeling process into bone and bone marrow, which we have extensively described in our previous publication (Pigeot et al., Advanced Materials 2021). However, the low amount of BMP-2 (in the dozens of nanogram/tissue range) embedded in the matrix is not sufficient per se to induce ectopic endochondral ossification. It is the combined presence of GAGs in the matrix -thus cartilage- that allows the success of bone formation.

- There is a great deal of missing detail in the manuscript.

We have incorporated additional methodological details describing the lyophilization/decellularization process of our tissues prior to evaluation (see Material and Methods section). We also have included a description of the MSOD-B line and implemented genetic elements (Supplementary Figure 1A).

- The in vivo study is underpowered, the results are not well documented pictorially, and are not convincing.

We believe our group size supports our conclusions confirmed by statistical assessment. We have provided additional stainings and images of higher magnifications (Figure 5) for both the ectopic and orthotopic in vivo evaluation.

- Given the fact that they have genetically modified cells, they could have done analyses of ECM components to determine what was different between the lines, both at the transcriptome and the protein level. Consequently, the study is purely descriptive and does not provide any mechanistic understanding of what mixture of matrix components and growth factors works best for cartilage or bone. But this presupposes that they actually induced the formation of bona fide cartilage, at least.

We thank the Reviewer for the suggestion. However, our study did not aim at understanding what ECM graft composition work best for cartilage nor bone regeneration respectively. Instead, we propose the exploitation of our cellular tools to interrogate the function of key ECM constituents and their impact in skeletal regeneration. We once more confirm that we generated cartilage grafts which is now better supported by additional histological assessment before lyophilization.